# Confidence predicts speed-accuracy tradeoff for subsequent decisions

Kobe Desender[1,2]*, Annika Boldt[3], Tom Verguts[2], Tobias H Donner[1]

[1]Department of Neurophysiology and Pathophysiology, University Medical Center, Hamburg, Germany; [2]Department of Experimental Psychology, Ghent University, Ghent, Belgium; [3]Institute of Cognitive Neuroscience, University College London, London, United Kingdom

**Abstract** When external feedback about decision outcomes is lacking, agents need to adapt their decision policies based on an internal estimate of the correctness of their choices (i.e., decision confidence). We hypothesized that agents use confidence to continuously update the tradeoff between the speed and accuracy of their decisions: When confidence is low in one decision, the agent needs more evidence before committing to a choice in the next decision, leading to slower but more accurate decisions. We tested this hypothesis by fitting a bounded accumulation decision model to behavioral data from three different perceptual choice tasks. Decision bounds indeed depended on the reported confidence on the previous trial, independent of objective accuracy. This increase in decision bound was predicted by a centro-parietal EEG component sensitive to confidence. We conclude that internally computed neural signals of confidence predict the ongoing adjustment of decision policies.

DOI: https://doi.org/10.7554/eLife.43499.001

## Introduction

Every day humans have to make numerous choices. These range from small and trivial (which shirt to wear) to complex and important (which house to buy). Such decisions are often based on ambiguous or noisy information about the state of the world. Human decision-makers are remarkably good at estimating their own accuracy, commonly reporting higher confidence for correct than for incorrect choices. Decision confidence can be conceptualized as the probability of a choice being correct, given the available evidence (*Pouget et al., 2016*; *Sanders et al., 2016*; *Urai et al., 2017*). A number of studies has investigated neural correlates of decision confidence (*Fleming et al., 2010*; *Kepecs et al., 2008*; *Kiani and Shadlen, 2009*), including attempts to dissociate subjective reports of decision confidence from objective decision accuracy (*Desender et al., 2018*; *Odegaard et al., 2018*; *Zylberberg et al., 2012*). An important open question is whether and how this subjective 'sense of confidence' is used to regulate subsequent behavior (*Meyniel et al., 2015*; *Yeung and Summerfield, 2012*). Theoretical treatments posit that, when information is sampled sequentially, confidence can be used to regulate how much information should be sampled before committing to a choice (*Meyniel et al., 2015*).

Here, we tested this prediction within the context of bounded accumulation models of decision-making (see *Figure 1* for illustration). Such models for two-choice tasks postulate the temporal accumulation of noisy sensory evidence towards one of two decision bounds, crossing of which determines commitment to one alternative (*Bogacz et al., 2006*; *Gold and Shadlen, 2007*; *Usher and McClelland, 2001*). The drift diffusion model (DDM; *Ratcliff and McKoon, 2008*) is a widely used instance of these models. Here, the mean drift rate quantifies the efficiency of evidence accumulation. Large signal-to-noise ratio (SNR) of the sensory evidence yields a large drift rate and, consequently, high accuracy and rapid decisions (conversely for low SNR; see *Figure 1A*). When evidence

**\*For correspondence:**
kobe.desender@gmail.com

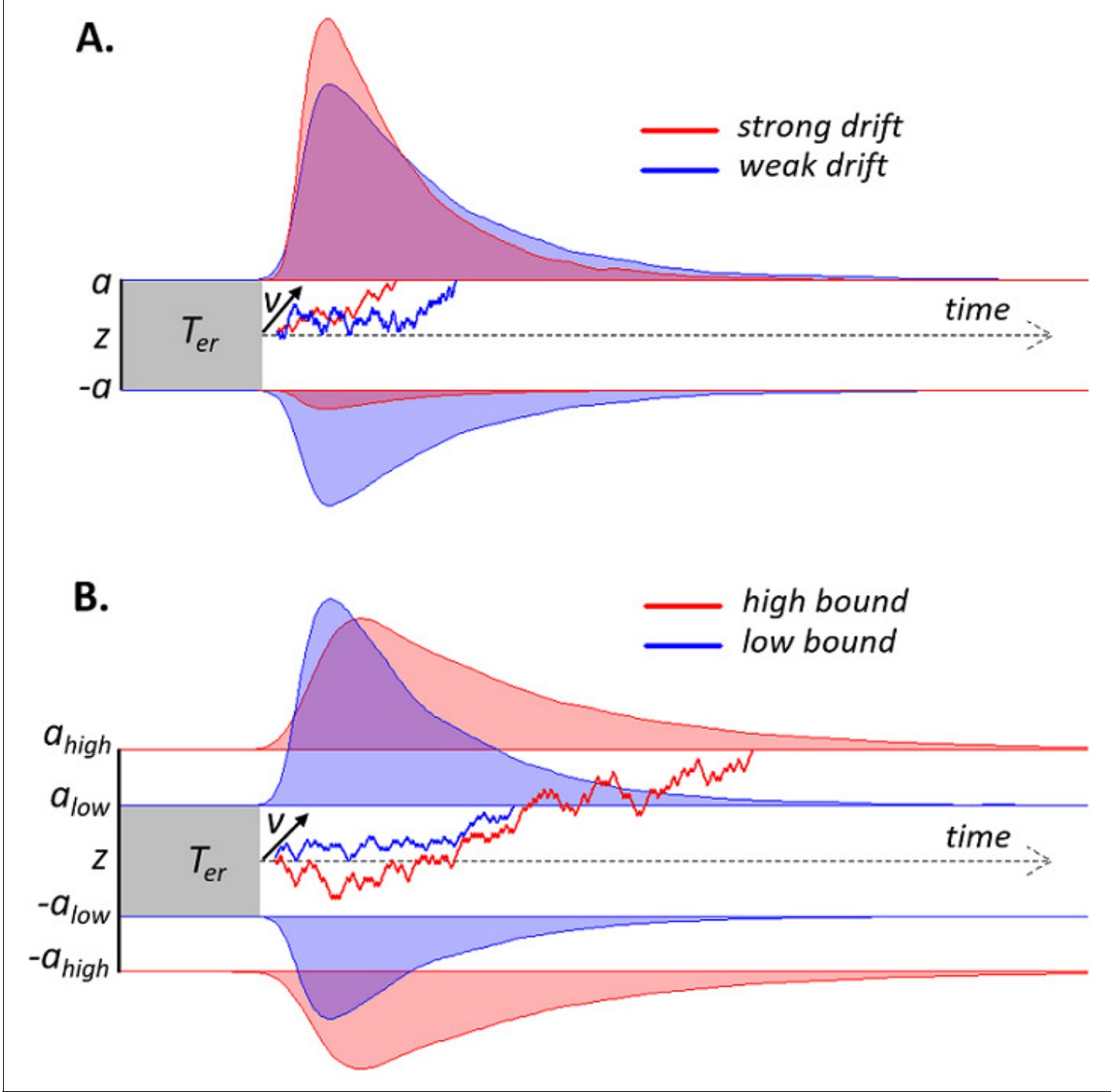

**Figure 1.** Schematic of drift diffusion model (DDM) with varying drift rates. (**A**) and varying decision bounds (**B**). Noisy sensory evidence is accumulated over time, until the decision variable reaches one of two bounds (a or -a), corresponding to correct and incorrect choices. The efficiency of information accumulation is given by v (mean drift rate). The time for sensory encoding and response execution is given by $T_{er}$. By increasing the separation between decision bounds, the probability of being correct increases, at the expense of prolonged reaction times. RT distributions (upper bounds) and error distributions (lower bounds) are depicted for different levels of drift rate (**A**) and decision bound (**B**).

DOI: https://doi.org/10.7554/eLife.43499.002

SNR is constant over trials, this sequential sampling process achieves an intended level of decision accuracy with the shortest decision time, or, conversely, an intended decision time at the highest accuracy level (*Gold and Shadlen, 2007*; *Moran, 2015*). The separation of the two decision bounds determines response caution, that is, the tradeoff between decision time and accuracy: The larger the bound separation, the more evidence is required before committing to a choice, increasing accuracy at the cost of slower decisions (*Figure 1B*). Controlling this decision bound separation, therefore, enables the decision-maker to prioritize either speed or accuracy (*Bogacz et al., 2010b*).

Several studies have shown that decision-makers can change their decision bounds as a function of external manipulations. For example, instructions to adhere to a liberal or conservative response strategy (*Forstmann et al., 2008*; *Hanks et al., 2014*; *Palmer et al., 2005*), or environments that reward fast or accurate responses (*Bogacz et al., 2010a*) all lead to changes in bound separation. Such manipulations typically rely on providing external instructions, or feedback, to the agent. In real-life decisions, explicit feedback about choice outcomes is often delayed or withheld. We hypothesized that, in the absence of external feedback, decision-makers set their decision bounds depending on *internal signals* encoding their decision confidence: in this way, low confidence about one choice gives rise to more cautious decision-making in the next (*Meyniel et al., 2015*; *Yeung and Summerfield, 2012*).

We found that decision confidence predicted decision bounds on the subsequent trial across three different perceptual choice tasks. A centro-parietal EEG component that tracked confidence on the current trial was linearly related to the subsequent-trial decision bound.

## Results

### Human confidence ratings exhibit signatures of statistical decision confidence

Twenty-eight human participants performed a task that has been used widely in computational and neurophysiological analyses of perceptual decision-making: discrimination of the net motion direction in dynamic random dot displays (*Bogacz et al., 2006*; *Gold and Shadlen, 2007*; *Siegel et al., 2011*). We asked participants to decide, as fast and accurate as possible, whether a subset of dots was moving coherently towards the left or right side of the screen. Decision difficulty was manipulated by varying the proportion of coherently moving dots. There were five different levels of coherence, ranging from 0 up to. 4, that were randomly intermixed in each block. After their choice, a 1 s blank screen or 1 s of continued motion (same coherence and direction as the initial stimulus) was shown, so as to allow for post-decisional accumulation of extra evidence, either from the sensory buffer (in the blank condition; *Resulaj et al., 2009*) or from the external stimulus (continued motion condition; *Fleming et al., 2018*). After this additional second, participants indicated how confident they felt about having made the correct choice (see *Figure 2A*).

As expected, RTs on correct trials and choice accuracy scaled with motion coherence level (*Figure 2B*; RTs: $F(4, 45.33)=30.61$, p<0.001, error rates: $X^2(4)=1285.6$, p<0.001). Correspondingly, drift rates estimated from DDM fits (see Materials and methods) also increased monotonically with coherence level (*Figure 2B*; Friedman $\chi2(5)=140$, p<0.001). In these model fits, decision bound separation was not allowed to vary as a function of coherence; its average estimate across participants was 2.09 ($SD = 0.33$). Similarly, non-decision time was held constant across levels of coherence; its average was 0.38 ($SD = 0.09$). Model fits closely captured the patterns seen in behavior (i.e., green crosses in *Figure 2B*), indicating that the DDM fitted the behavioral data well.

Participants' confidence ratings exhibited a key signature of statistical decision confidence established in previous work (*Sanders et al., 2016*; *Urai et al., 2017*): an opposite-sign relation between evidence strength and confidence for correct and incorrect choices (*Figure 2C*). The scaling of confidence judgments with coherence level ($F(4,52.6) = 4.56$, p=0.003) depended on choice accuracy ($F(4,6824.6) = 154.52$, p<0.001), with confidence increasing with coherence levels for correct trials (linear contrast: p<0.001) and decreasing for error trials (linear contrast: p<0.001). This pattern was highly similar in blocks with continued evidence following the choice and blocks in which choices were followed by a blank screen (see *Figure 2—figure supplement 1*). A Bayesian ANOVA confirmed that the interactive effects of coherence and accuracy on confidence were similar for both conditions, $BF = 0.05$ (i.e., the null hypothesis was 20 times more likely than the alternative).

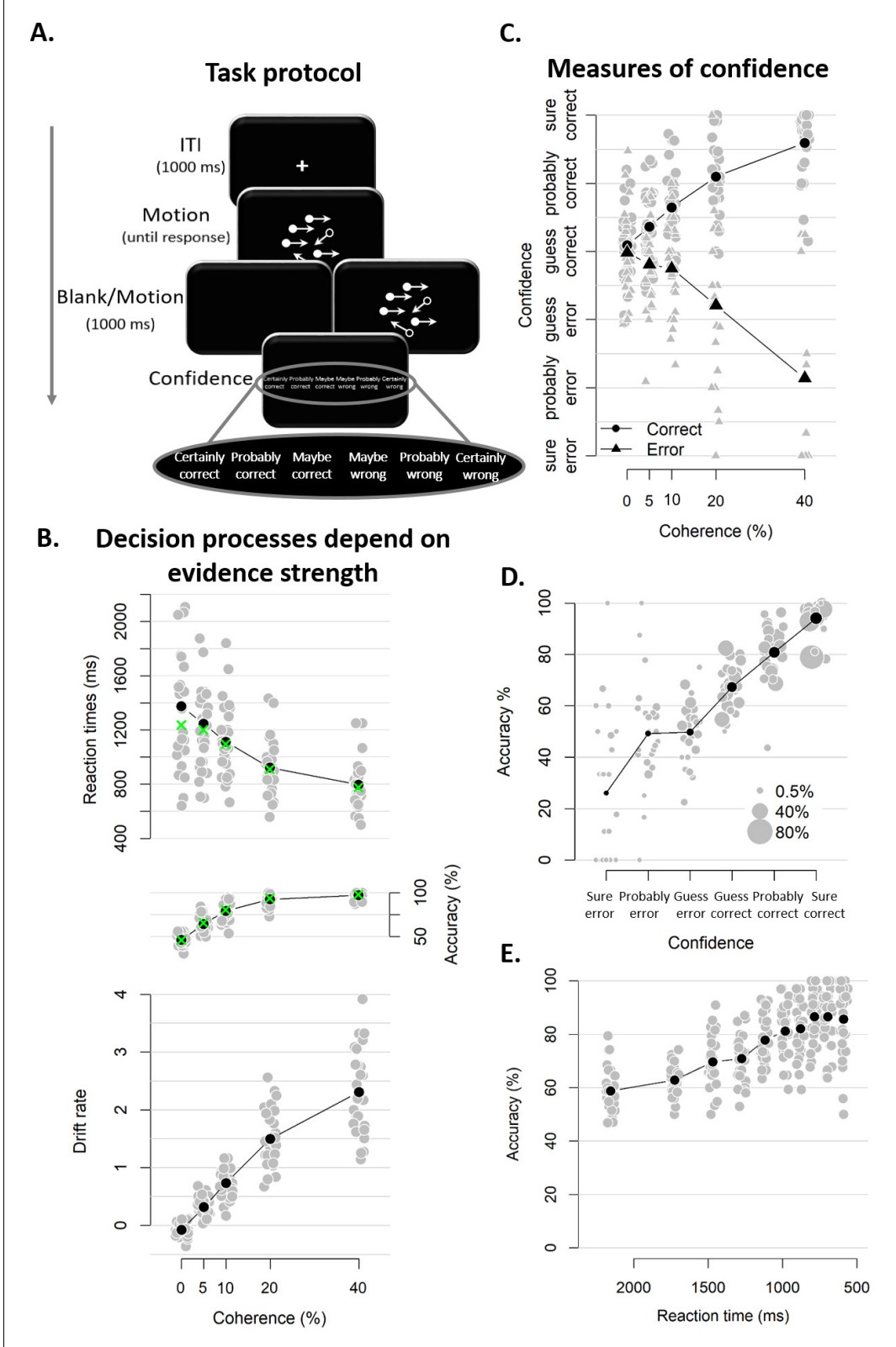

**Figure 2.** Experimental task and results from Experiment 1. (**A**) Sequence of events in a trial from Experiment 1. Participants decided, as fast and accurately as possible, whether the majority of dots were moving left or right. After their response and a 1 s blank or 1 s of continued motion, they indicated the degree of confidence in their decision using a six-point confidence scale (ranging from certainly correct to certainly wrong). (**B**) Mean reaction time on correct trials (top), accuracy (middle) and estimated mean drift rate (bottom) as a function of coherence. Green crosses show fits from
*Figure 2 continued on next page*

*Figure 2 continued*

the DDM. (C) Confidence as a function of coherence level, separately for corrects and errors. (D) Accuracy as a function of decision confidence (dot size reflects percentage of trials per confidence label, separately for each participant). (E) Accuracy as a function of reaction time. Data are pooled across the blank and the continued motion condition. Gray dots: individual participants; black dots: group averages.

DOI: https://doi.org/10.7554/eLife.43499.003

The following figure supplement is available for figure 2:

**Figure supplement 1.** Similar confidence judgments for blocks with and without post-decisional evidence.

DOI: https://doi.org/10.7554/eLife.43499.004

Correspondingly, confidence ratings were closely linked to choice accuracy (*Figure 2D*). Confidence ratings were monotonically related to choice accuracy on a trial-by-trial basis, even after factoring out motion coherence (logistic regression of confidence on accuracy with coherence as a covariate: positive slopes for all observers, 23 *p*s <0.025, five non-significant).

Notably, and different from previous work (*Sanders et al., 2016*), confidence ratings predicted accuracy from below maximum uncertainty (i.e., 50%) to about 100%: Rating-predicted accuracy ranged from 23% (certain error) up to 94% (certain correct), both significantly different from chance level (p<0.001). By contrast, RTs for the initial choice (pooled across difficulty levels), while also monotonically related to accuracy (*b* = −0.03, *t*(27) = −9.41, p<0.001) predicted accuracy variations only from about 60% to 90% correct (*Figure 2E*), similar to previous results (*Sanders et al., 2016*; *Urai et al., 2017*). This shows that human confidence ratings can lawfully account for certainty about errors (i.e., accuracy levels below 50%) when such certainty is enabled by the experimental protocol, due to post-decisional evidence accumulation (see also *Fleming et al., 2018*). This generalizes the signatures of decision confidence (as defined above) reported by previous analyses of reaction times or confidence reports (*Sanders et al., 2016*; *Urai et al., 2017*) to the domain of error detection. We next sought to pinpoint the consequences of trial-to-trial variations in participants' confidence ratings on the subsequent decision process.

## Decision confidence influences subsequent decision bound

We hypothesized that response caution would increase after low-confidence decisions (in particular after 'perceived errors'), a change in speed-accuracy tradeoff mediated by an increase in decision bound. A bound increase will increase both RT and accuracy, a trend that was evident in the data (*Figure 3A*, left). We multiplied median RT and mean accuracy, separately for each level of confidence, to combine both effects into a single, model-free measure of response caution (*Figure 3A*, right). This aggregate measure of response caution was predicted by the confidence rating from the previous decision, *F*(2,81) = 3.13, p=0.049. Post-hoc contrasts showed increased caution after perceived errors compared to after both high confidence, *z* = 2.27, p=0.032, and low confidence, *z* = 2.05, p=0.041. There was no difference in caution following high and low confidence ratings, p=0.823.

The confidence-dependent change in subsequent response caution was explained by a DDM, in which decision bounds and drift rate could vary as a function of previous confidence (*Figure 3A*, green crosses; see Materials and methods). We used a hierarchical DDM regression approach to fit this model (*Wiecki et al., 2013*), see Materials and methods. In this fitting approach, parameter estimates for individual participants are constrained by the group prior, whereby the contribution of each individual to the group prior depends on the number of trials available from that participant in the corresponding condition (Materials and methods). One important consequence of this approach is that individual parameter estimates are not independent and statistical inferences are only meaningful at the group level. We treated high confidence trials as reference category in the regression, so parameter values reflect deviations (i.e. delta scores) from the parameter estimate for high confidence. All summary statistics of the observed data fell within the 95% credibility interval of the fitted RTs.

As predicted, the subsequent-trial separation of decision bounds scaled monotonically with the complement of decision confidence (i.e., uncertainty; *Figure 3B*). Subsequent decision bound increased after low compared to high confidence decisions (*M* = 0.083, *SD* = 0.047, p=0.037) and even further after participants perceived an error (*M* = 0.262, *SD* = 0.078, p<0.001). Decision bound

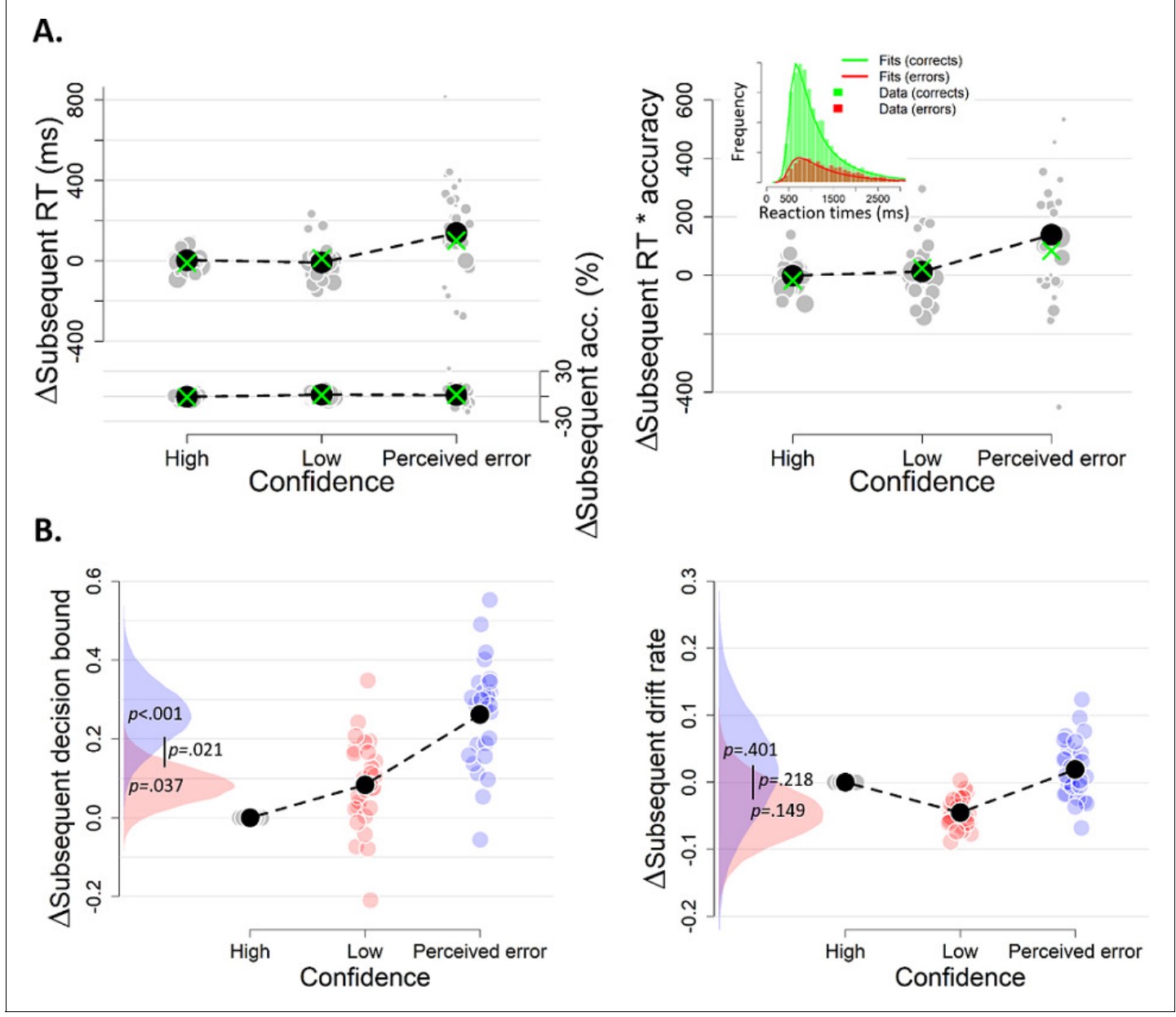

**Figure 3.** The influence of decision confidence on subsequent decision bound. (**A**) Model-free measures of response caution on trial n+one as function of confidence on trial n: mean RT, accuracy, and their product. Inset, distribution of empirical and fitted RTs. (**B**) Model-based estimate of decision bound and drift rates on trial n+one as function of confidence on trial n. Distributions show the group posteriors over parameter estimates. In all panels, as well as in all subsequent figures, 'delta' on y-axis refers to deviation of dependent variable from its value in the high-confidence condition (i. e., centered on zero for high-confidence). For model fits, trials from the high-confidence condition served as reference so that parameter estimates reflected deviations from their value on high-confidence. Statistical significance is reflected in overlap between posterior distributions over parameter estimates (Materials and methods).

DOI: https://doi.org/10.7554/eLife.43499.005

The following figure supplements are available for figure 3:

**Figure supplement 1.** The influence of decision confidence on subsequent decision bounds and drift rates, separately for blocks with post-decisional evidence presentation (**A–B**) and a post-decisional blank (**C–D**).

DOI: https://doi.org/10.7554/eLife.43499.006

**Figure supplement 2.** Simple effects of confidence on trial$_n$ and confidence on trial$_{n+2}$ on the product of subsequent RTs and accuracy as a model-free measure of decision bound (Experiment 1).

DOI: https://doi.org/10.7554/eLife.43499.007

*Figure 3 continued on next page*

*Figure 3 continued*

**Figure supplement 3.** Simple effects of confidence on trial$_n$ and confidence on trial$_{n+2}$ on decision bound (**A**) and drift rate (**B**) on trial$_{n+1}$ (Experiment 1).

DOI: https://doi.org/10.7554/eLife.43499.008

**Figure supplement 4.** Complementary approach controlling for slow drifts in performance (Experiment 1).

DOI: https://doi.org/10.7554/eLife.43499.009

after perceived errors was also larger compared to after low confidence decisions (p=0.021). The posterior distribution for subsequent decision bounds for perceived errors (in blue) overlapped only slightly with the distribution for low confidence trials (in red), and barely overlapped with that for high confidence trials (zero; the reference category, *Figure 3B*). The *p*-values presented in the figure directly reflect this overlap. This pattern of results was similar in blocks with and blocks without post-decisional evidence presentation (see *Figure 3—figure supplement 1*). In sum, these results demonstrate that decision bounds are increased following perceived errors. Decision confidence had no effect on subsequent drift rate (*Figure 3B*).

The results in *Figure 3* were obtained by fitting the regression model as function of current-trial confidence, ignoring current-trial choice accuracy. So, for example, trials labeled as 'perceived errors' contained a mixture of correct and error trials. Part of the results could thus reflect previously established effects of post-error slowing in decision-making (*Purcell and Kiani, 2016*). When estimating the effects of current confidence separately for current correct and error trials, we found that the confidence rating-dependence of subsequent decision bound holds even for variations of confidence ratings within both correct (*Figure 4A*) and error trials (*Figure 4B*). This result shows that the modulation of decision bound is specifically due to trial-by-trial variations in internal confidence signals, rather than the objective accuracy of the choice, thus going beyond the previous findings.

The effects of confidence ratings on subsequent decision bounds are unlikely to be caused by our systematic manipulation of evidence strength (i.e., motion coherence) or evidence volatility (Materials and methods). Confidence ratings were reliably affected by both evidence strength (as reported before; *Figure 2C*) and evidence volatility (data not shown, $F(1, 26.7)=47.10$, p<0.001, reflecting higher confidence with high evidence volatility). However, evidence strength and volatility, in turn, did not affect subsequent decision bound, both $p$s > 0.133. For each of these, zero (i.e., no effect) was included in the 95% highest density interval of the posterior distribution ($-0.167$ to. 026, and $-0.020$ to. 039, respectively), suggesting it is likely that the true parameter value was close to zero.

The analyses on the behavioral measure of response caution and on the model fits (shown in *Figure 3*) control for slow drifts in performance over the course of the experiment (*Dutilh et al., 2012b*; *Gilden, 2003*; *Palva et al., 2013*), which likely reflect slow and non-specific fluctuations in behavioral state factors (e.g. motivation, arousal, attention). Indeed, slow ('scale-free') fluctuations similar to those reported previously (*Gilden, 2003*; *Palva et al., 2013*) were present in the current RT and confidence rating (slopes of linear fits to log-log spectra of the corresponding time series were significant for both RTs, $b = -0.42$, $t(23) = -11.21$, p<0.001, and confidence, $b = -0.60$, $t(23) = -13.15$, p<0.001). To appreciate this result, consider a streak of trials during which arousal level declines monotonically. This will cause a monotonic increase in the 'noise' of sensory responses (*McGinley et al., 2015*; *Reimer et al., 2014*), which, in turn, will translate into a monotonic decrease of drift, decision accuracy, and decision confidence (*Kepecs et al., 2008*), accompanied by an increase in RT. Such effects would cause changes in all behavioral variables and DDM parameter estimates from trial n to trial n+1, without reflecting the rapid and strategic, confidence-dependent adjustments of decision policy, which were of interest for the current study. We hypothesized that latter adjustments were superimposed onto the former, slow performance drifts. To isolate the confidence-dependent trial-by-trial adjustments, we removed the influence of slow performance drifts: we subtracted the effect of decision confidence on trial$_{n+2}$ on the dependent variables on trial$_{n+1}$ from the effect of confidence on trial$_n$ in *Figure 3* (see Materials and methods and *Figure 3—figure supplement 2* and *Figure 3—figure supplement 3* for the 'raw' effects for trial$_n$ and trial$_{n+2}$).

A possible concern is that the decision bound on trial$_{n+1}$ was correlated with confidence ratings on trial$_{n+2}$, which would confound our measure of the effect of confidence on trial$_n$ on decision bound on trial$_{n+1}$. Two observations indicate that this does not explain our findings. First, the observed association between confidence ratings on trial$_n$ and decision bound on trial$_{n+1}$ was also

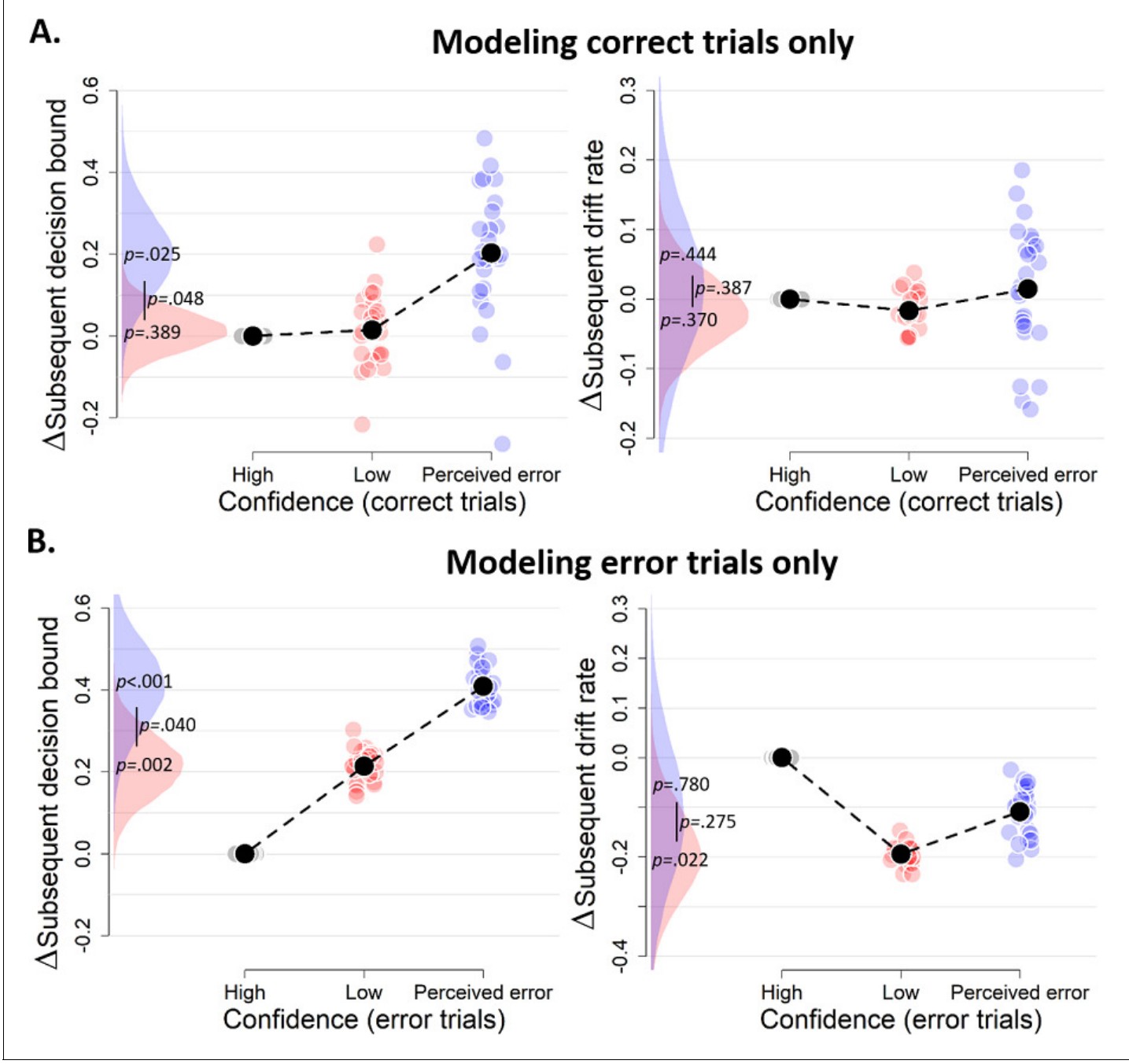

**Figure 4.** Confidence-dependent modulation of decision bound remains qualitatively similar when modeling only correct (**A**) or only error trials (**B**). Same conventions as in *Figure 3*. Note that, due to a lack of trials in one of the cells, results are based on 25 (**A**) and 24 (**B**) participants.
DOI: https://doi.org/10.7554/eLife.43499.010

evident in the 'raw' parameter values for the bound modulation, that is, without removing the effects of slow performance drift (*Figure 3—figure supplement 3*). Second, when using a complementary approach adopted from the post-error slowing literature (*Dutilh et al., 2012b*; see Materials and methods), we observed largely similar results (see *Figure 3—figure supplement 4*).

## Confidence-dependent modulation of decision bound generalizes to other tasks

Having established a robust effect of confidence ratings on subsequent response caution and decision bound in the dot motion discrimination task, we tested for the generalization of the effect to other perceptual choice tasks. First, we reanalyzed previously published data (*Boldt and Yeung, 2015*) from an experiment in which sixteen participants performed a speeded decision task, in which they decided as quickly as possible which of two boxes contained more dots (Experiment 2; *Figure 5A*, left). Different from Experiment 1, in this dataset only a single level of difficulty was used, thus allowing us to test whether the findings of Experiment one generalize to internal variations of confidence occurring at a fixed evidence SNR. Similar to Experiment 1, both RTs and confidence judgments predicted choice accuracy (see *Figure 5—figure supplement 1*). As in Experiment 1, our model-free measure of response caution (RT*accuracy) was modulated by confidence ratings on the previous trial, $F(2,45) = 3.21$, p=0.050 (perceived errors vs. high confidence: $z = 2.53$, p=0.011; no significant differences for other comparisons $ps$ >0.178; *Figure 5B*; see *Figure 5—figure supplement 2A* for the 'raw' effects of trial$_n$ and trial$_{n+2}$).

Second, we analyzed data from an experiment, in which twenty-three participants performed a visual color categorization task, designed after *de Gardelle and Summerfield (2011)*, deciding as fast as possible whether the mean color of eight elements was red or blue (Experiment 3; *Figure 5A*, right). Task difficulty was manipulated by independently varying the distance of the color mean from the category bound and the standard deviation across the elements' colors. Both variables together determined the SNR of the sensory evidence (i.e., mean distance from category boundary/variance). Similar to Experiment 1, RTs, accuracy and drift rate scaled monotonically with SNR, and both RTs and confidence judgments predicted choice accuracy (see *Figure 5—figure supplement 5*). Our model-free measure of response caution was again affected by previous confidence ratings, $F(2,66) = 14.43$, p<0.001 (perceived errors vs. high confidence: $z = 4.61$, p<0.001; perceived errors vs. low confidence: $z = 4.69$, p<0.001; high vs. low confidence: p=0.938; *Figure 5B*; see *Figure 5—figure supplement 2B* for the 'raw' effects of trial$_n$ and trial$_{n+2}$).

Also for Experiments 2 and 3, the modulation of behavior by confidence was captured by the DDM fits to the data (*Figure 5B*; green crosses). All summary statistics of the observed data fell within the 95% credibility interval of the fitted RTs. In both datasets, we again found that subsequent decision bounds were modulated by decision confidence (see *Figure 5C*). When participants perceived to have committed an error, subsequent decision bounds were increased (Exp2: $M = 0.110$, $SD = 0.046$; Exp3: $M = 0.117$, $SD = 0.046$) compared to having high confidence (Exp2: p=0.007; Exp3: p=0.004) or low confidence (Exp 2: $M = 0.059$, $SD = 0.038$, p=0.059; Exp 3: $M = -0.046$, $SD = 0.027$, p<0.001). In Experiment 3, subsequent decision bounds were unexpectedly lower following low confidence trials compared to high confidence (p=0.043). Again, the effects of confidence ratings on subsequent decision bounds were present separately for confidence ratings on correct (*Figure 6A*) and error trials (*Figure 6B*). As in Experiment 1, the systematic trial-to-trial variations of evidence strength (SNR) did not influence subsequent decision bound in Experiment 3 (p=0.220), and zero was included in the 95% highest density interval of the posterior (−0.010 to. 031). Finally, we again observed a robust effect of confidence ratings on subsequent decision bound when using the above described alternative procedure to control for slow performance drift (*Figure 5—figure supplement 4* and *Figure 5—figure supplement 7*). In Experiment 3 (*Figure 5—figure supplement 6*) but not in Experiment 2 (*Figure 5—figure supplement 3*), this effect was also present without controlling for slow performance drifts.

Both datasets also showed a small modulation of subsequent drift rate by decision confidence, an effect not present in Experiment 1, but consistent with recent studies of post error slowing (*Purcell and Kiani, 2016*): When participants had low confidence in their choice, mean drift rate on the subsequent trial was lower (Exp2: $M = -0.216$, $SD = 0.121$; Exp3: $M = -0.149$, $SD = 0.085$) relative to high confidence (Exp2: p=0.039; Exp3: p=0.039) and trials perceived as errors (Exp2: p=0.122; Exp3: p=0.034). The latter two were not different (Exp2: p=0.378; Exp3: p=0.169).

## A neural marker of confidence predicts subsequent decision bound

The results from the previous sections indicated that confidence modulates the separation of the bounds for subsequent decisions. Which internal signals are used to transform confidence estimates

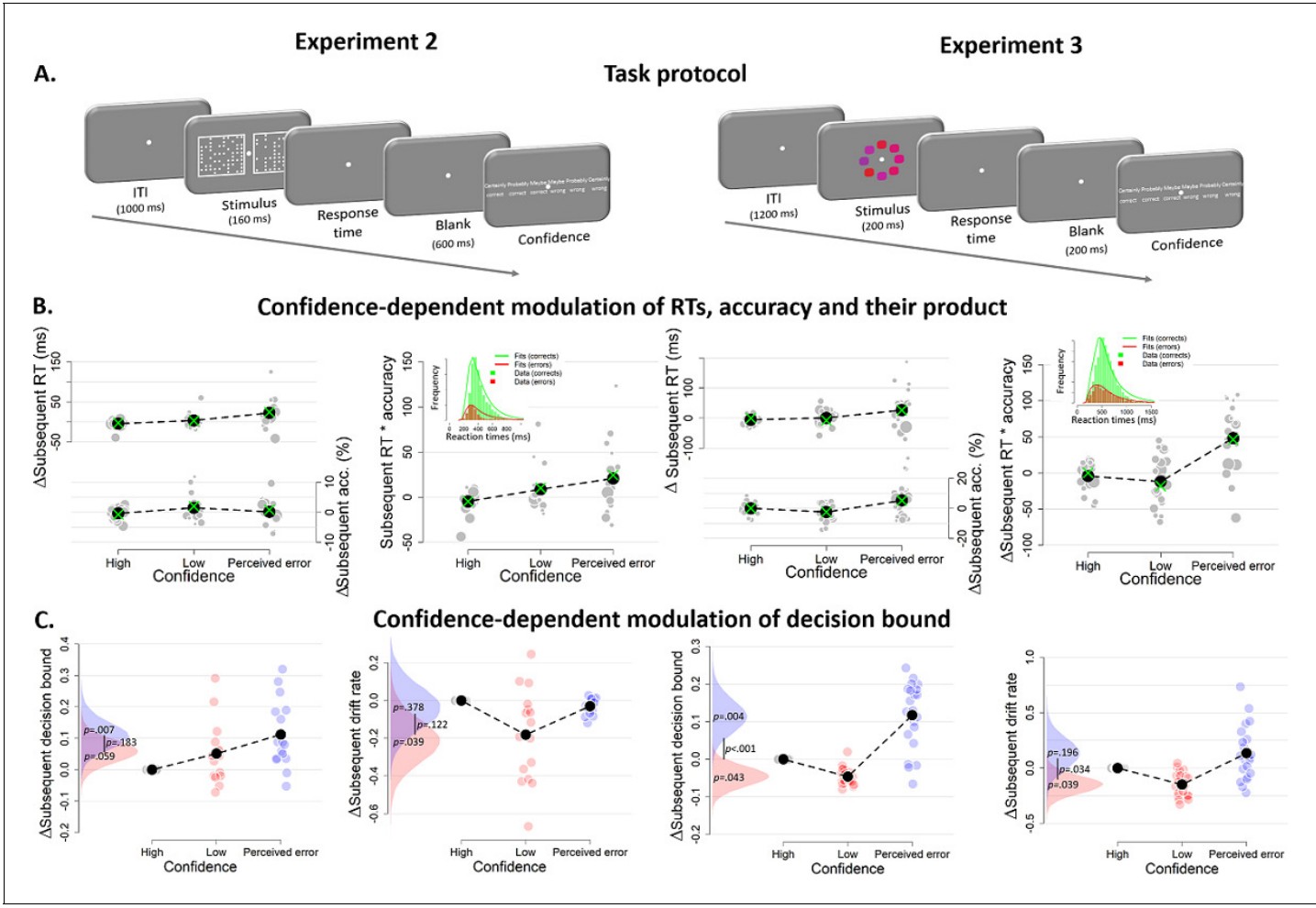

**Figure 5.** Confidence-dependent modulation of decision bounds generalizes to different experimental tasks. (**A**) In Experiment 2, participants need to decide as quickly as possible which of two boxes has more dots. In Experiment 3, participants needed to decide, as fast and accurately as possible, whether the average color of the eight elements was more red or blue. (**B**) Subsequent RT, accuracy and their product as a model-free measure of decision bound as a function of confidence. Green crosses show fits from the DDM. (**C**) Subsequent decision bounds and subsequent drift rates as a function of confidence. Inset in B shows the distribution of empirical and fitted RTs.

DOI: https://doi.org/10.7554/eLife.43499.011

The following figure supplements are available for figure 5:

**Figure supplement 1.** Behavioral results of Experiment 2.

DOI: https://doi.org/10.7554/eLife.43499.012

**Figure supplement 2.** Simple effects of confidence on trial$_n$ and confidence on trial$_{n+2}$ on the product of subsequent RTs and accuracy as a model-free measure of decision bound for Experiment 1 (**A**) and Experiment 2 (**B**).

DOI: https://doi.org/10.7554/eLife.43499.013

**Figure supplement 3.** Simple effects of confidence on trial$_n$ and confidence on trial$_{n+2}$ on decision bound (**A**) and drift rate (**B**) on trial$_{n+1}$ (Experiment 2).

DOI: https://doi.org/10.7554/eLife.43499.014

**Figure supplement 4.** Complementary approach controlling for slow fluctuations (Experiment 2).

DOI: https://doi.org/10.7554/eLife.43499.015

**Figure supplement 5.** Behavioral results of Experiment 3.

DOI: https://doi.org/10.7554/eLife.43499.016

**Figure supplement 6.** Simple effects of confidence on trial$_n$ and confidence on trial$_{n+2}$ on decision bound (**A**) and drift rate (**B**) on trial$_{n+1}$ (Experiment 3).

DOI: https://doi.org/10.7554/eLife.43499.017

**Figure supplement 7.** Complementary approach controlling for slow drifts in performance (Experiment 3).

DOI: https://doi.org/10.7554/eLife.43499.018

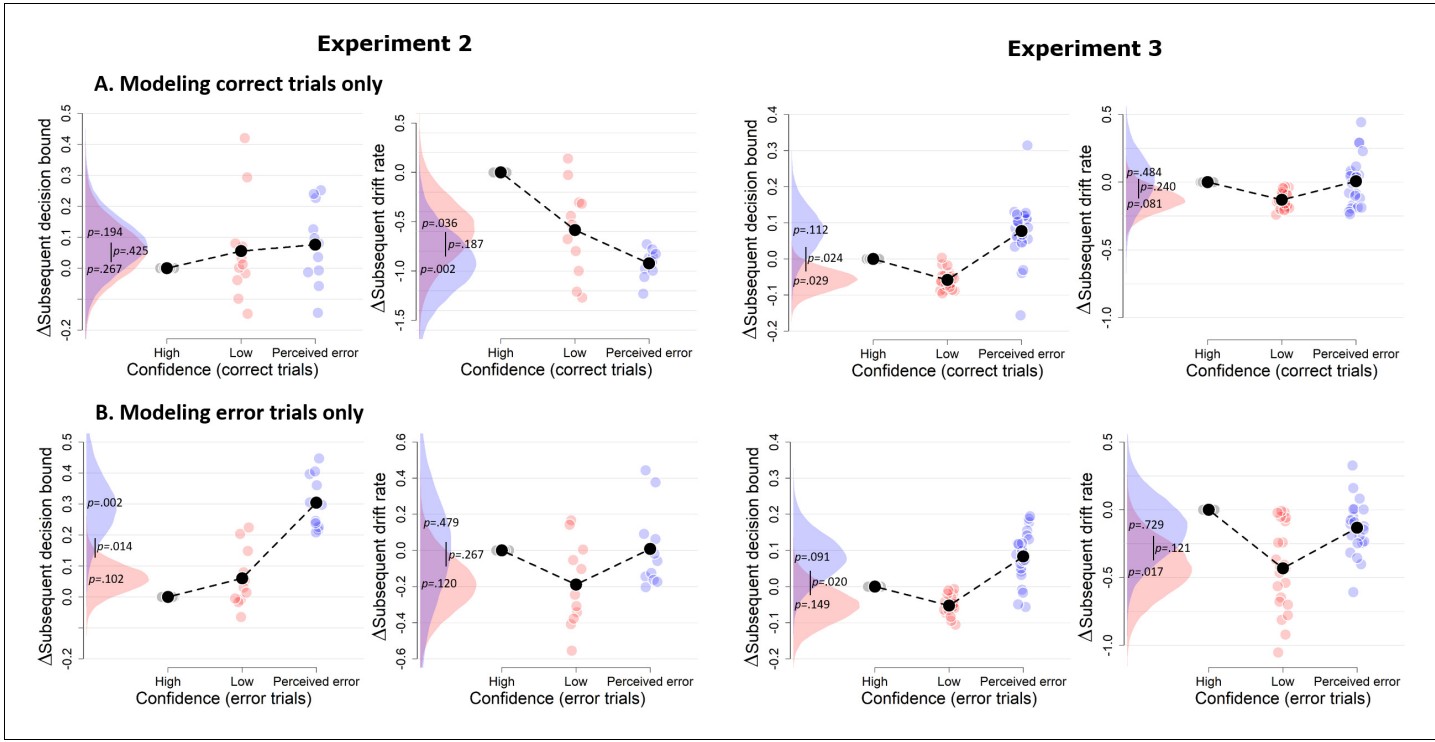

**Figure 6.** Confidence-dependent modulation of decision bound remains qualitatively similar when modeling only correct (**A**) or only error trials (**B**). Due to a lack of trials in one of the cells, the results shown in A and B are based on 11 participants for Experiment 2, and respectively 22 and 23 participants for Experiment 3.

DOI: https://doi.org/10.7554/eLife.43499.019

into changes in subsequent decision bound? Two components of the human EEG evoked potential are established neurophysiological markers of confidence and error processing: (i) the error-related negativitiy (ERN), a fronto-central signal peaking around the time of the response; and (ii) the Pe, a centro-parietal signal that follows the ERN in time. The ERN originates from mid-frontal cortex (*Dehaene et al., 1994*; *Van Veen and Carter, 2002*) and has been implicated in error processing. Different accounts postulate that the ERN reflects a mismatch between the intended and the actual response (*Charles et al., 2013*; *Nieuwenhuis et al., 2001*), the detection of conflict (*Yeung et al., 2004*), or a negative prediction error (*Holroyd and Coles, 2002*). The Pe was initially linked to error perception (hence its name; *Nieuwenhuis et al., 2001*) and more recently to post-decisional evidence accumulation (*Murphy et al., 2015*) as well as fine-grained variations in decision confidence (*Boldt and Yeung, 2015*).

We here used the EEG data that were collected in Experiment two to test if the confidence-dependent modulation of subsequent decision bound was linked to one, or both, of these confidence-related neural signals. We reasoned that these neural data may provide a more veridical measure of the internal confidence signals governing subsequent behavior than the overt ratings provided by participants, which require additional transformations (and thus additional noise sources) and are likely biased by inter-individual differences in scale use and calibration. Furthermore, quantifying the unique contribution of both confidence-related neural signals to bound adjustment allowed for testing for the specificity of their functional roles, an important issue given their distinct latencies and neuroanatomical sources.

Both the Pe and the ERN were modulated by decision confidence (*Figure 7A*), as already shown in the original report of these data (*Boldt and Yeung, 2015*). ERN amplitudes at electrode FCz were monotonically modulated by confidence, $F(2,14) = 18.89$, p<0.001, and all conditions differed from each other, |ts| > 3.62 and ps <0.003. Likewise, Pe amplitude at electrode Pz was monotonically modulated by confidence, $F(2,14) = 19.19$, p<0.001, and all conditions differed from each other, |ts| > 2.19, ps <0.045.

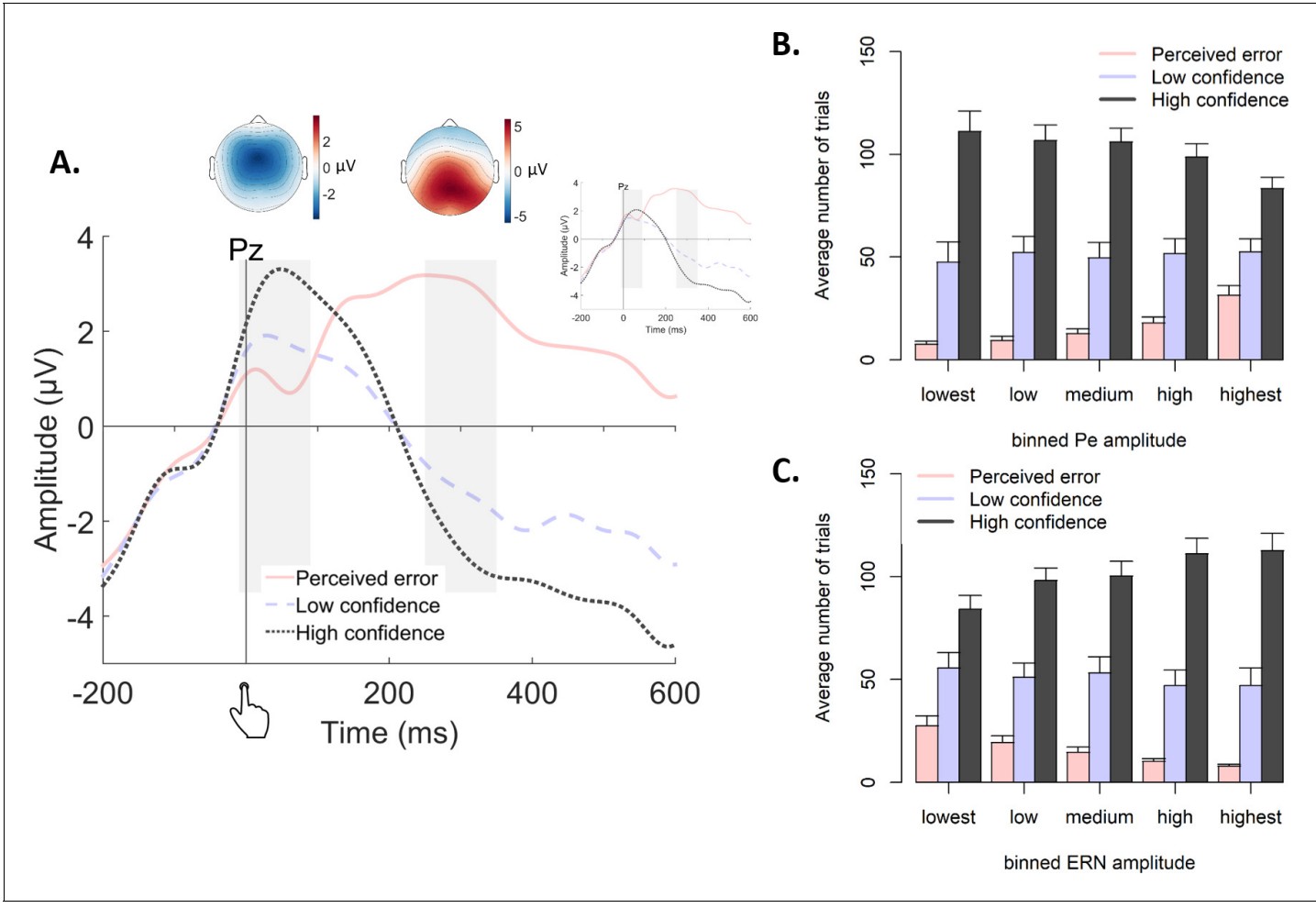

**Figure 7.** Post-decisional ERPs in Experiment 2. (A) Response-locked grand-average ERPs at electrode Pz, separately for the three levels of confidence. Gray bars represent the ERN (−10 ms to 90 ms) and the Pe (250 ms to 350 ms). Inset shows the same ERPs after regressing out the ERN. Topographic plots display amplitude differences between high confidence and perceived errors. (B–C) Average number of trials in each bin of the Pe (C) and the ERN (D), separately for the levels of confidence.

DOI: https://doi.org/10.7554/eLife.43499.020

We then fitted the DDM to the data with both EEG markers as (trial-to-trial) covariates (i.e., ignoring decision confidence) to test if either or both post-decisional EEG signals predicted subsequent decision bound (see *Figure 8A*). The regression coefficient relating the Pe to subsequent decision bound was significantly different from zero (p<0.001) whereas the regression coefficient for the ERN was not (p=0.327). Regression coefficients of the Pe and the ERN were significantly different from each other (p<0.001). The sign of the Pe regression coefficient was positive, indicating that more positive Pe amplitudes predicted an increase of subsequent decision bound. Concerning the drift rate, the coefficient relating the Pe to subsequent drift rate differed from zero, p<0.001, whereas the ERN was unrelated to subsequent drift rate, p=0.340. Regression coefficients of the Pe and ERN were significantly different, p<0.001. The sign of the Pe regression coefficient was negative, indicating that more positive Pe amplitudes were related to smaller drift rates on the subsequent trial. The isolated effects uncontrolled for trial$_{n+2}$ are shown in *Figure 8—figure supplement 1*.

## The error positivity (Pe) linearly scales with subsequent decision bound

The previous section showed that the Pe was related to subsequent decision bound and drift rate, but that analysis could not reveal potential nonlinearities in this relationship. We therefore divided the Pe into five equal-sized bins based on its amplitude, separately for each participant. In the

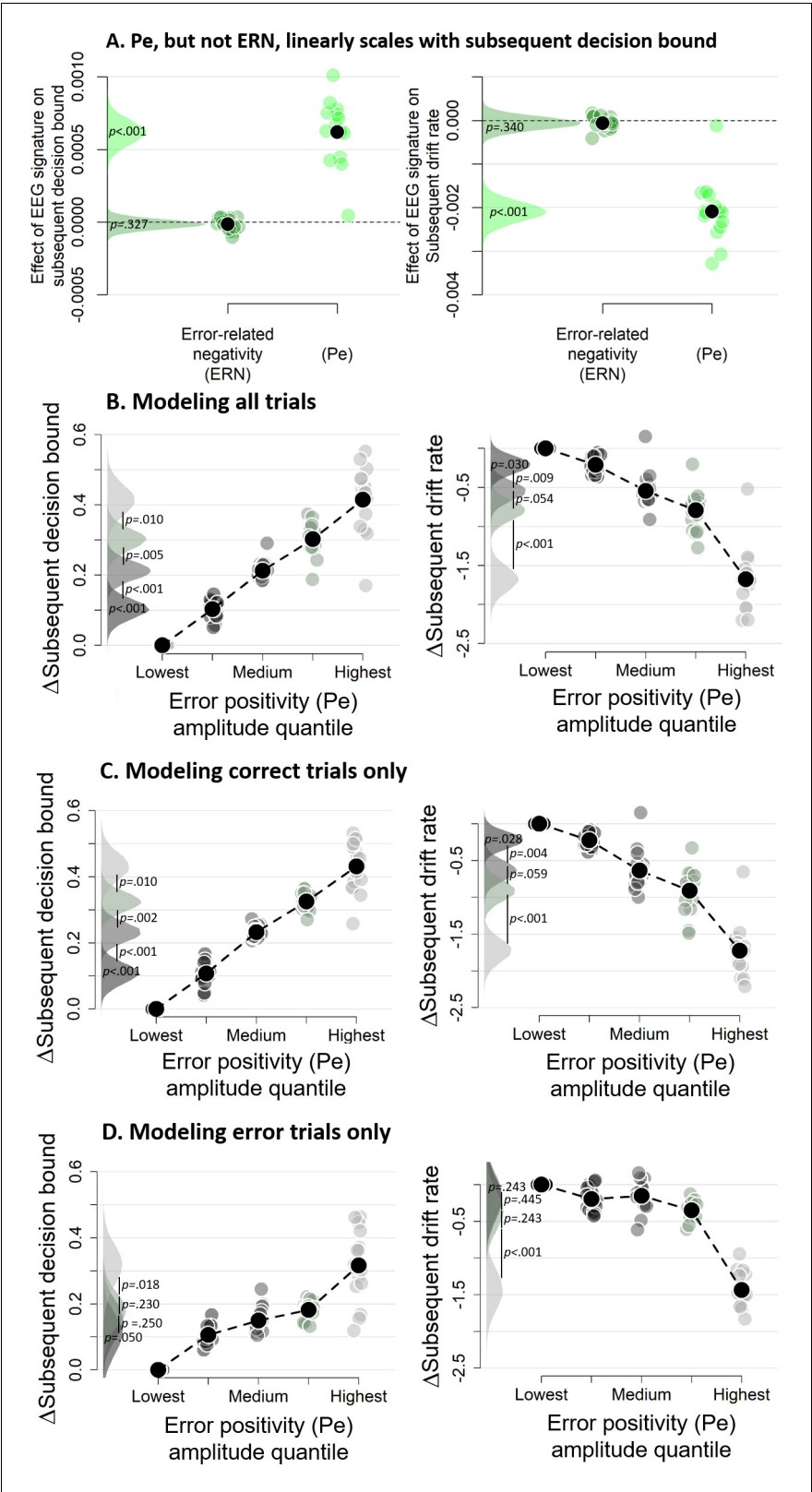

**Figure 8.** Subsequent decision bounds and drift rates (**A**) as a function of Pe and ERN. (**B**) Pe-dependent variations in subsequent decision computation. (**C-D**) Modeling results remain qualitatively similar when modeling only correct (**B**) or only error trials (**C**). The bin with the lowest Pe amplitude quantile was always treated as reference category (i.e., fixed to zero). Data were fit using the regression approach, so values reflect coefficients.

*Figure 8 continued on next page*

*Figure 8 continued*

DOI: https://doi.org/10.7554/eLife.43499.021

The following figure supplements are available for figure 8:

**Figure supplement 1.** Simple effects of rank-ordered Pe and ERN amplitude on trial$_n$ (A and C) and trial$_{n+2}$ (B and D) on decision bound (A and B) and drift rate (C and D) on trial$_{n+1}$.

DOI: https://doi.org/10.7554/eLife.43499.022

**Figure supplement 2.** Pe effects without regressing out the ERN.

DOI: https://doi.org/10.7554/eLife.43499.023

**Figure supplement 3.** Simple effects of binned Pe amplitude on trial n (A and C) and binned Pe amplitude trial$_{n+2}$ (B and D) on decision bound (A and B) and drift rate (C and D) on trial$_{n+1}$ (Experiment 2).

DOI: https://doi.org/10.7554/eLife.43499.024

**Figure supplement 4.** Complementary approach controlling for slow drifts in performance (EEG data).

DOI: https://doi.org/10.7554/eLife.43499.025

previous section, both components were entered in a single model, thus in that fit the regression coefficients capture unique variance of each signal. The trial-to-trial variations in Pe amplitudes were correlated with those of the ERN amplitudes, mean $r$ = 0.180, $t(15)$ = 6.92, p<0.001. In order to again capture the effect unique to the Pe, bins were created after the ERN was regressed out of the Pe, separately per participant, which did not affect the confidence scaling of the Pe (inset of *Figure 7A*). Note that the results described below remain largely unchanged when creating bins based on raw Pe amplitude (*Figure 8—figure supplement 2*). *Figure 7B–C* show the distribution of confidence judgments over the different bins.

Subsequent decision bounds increased monotonically (and approximately linearly) as a function of binned Pe amplitude quantile, Friedman χ2(4)=61.00, p<0.001 (*Figure 8B*), with all adjacent bins differing significantly from each other, all $ps$ < 0.010. Likewise, subsequent drift rates linearly decreased as a function of binned Pe amplitude quantile, Friedman χ2(4)=61.75, p<0.001, and all adjacent bins were significantly different from each other, all $ps$ < 0.055. The simple effects, uncontrolled for slow performance drifts, are shown in *Figure 8—figure supplement 3*. Similar findings were obtained using our alternative approach to control for slow performance drifts (*Figure 8—figure supplement 4*). Finally, fitting the same model selectively on correct trials (*Figure 8C*) or error trials (*Figure 8D*) provided highly similar results. In sum, there was an approximately linear relationship between the amplitude of the error positivity (Pe) and subsequent decision bound separation as well as subsequent drift rate. Thus, the Pe qualifies as a neural marker of decision confidence predicting flexible, trial-to-trial adaptation of the decision bounds.

## Discussion

Accumulation-to-bound models of decision making assume that choices are formed once the integration of noisy evidence reaches a bound. This decision bound is commonly assumed to be fixed within a block of constant external task conditions (*Ratcliff and McKoon, 2008*). Here, we show that this decision bound, in fact, dynamically changes from trial to trial, dependent on the confidence about the previous decision: In three independent datasets the separation between decision bounds increased after participants sensed they had made an error. Importantly, this was observed independent of the objective accuracy of a trial. A post-decisional brain signal, the so-called Pe component, scaled with decision confidence and linearly predicted the decision bound on the subsequent trial. These findings indicate that, in the absence of external feedback about choice outcome, decision-makers use internal confidence signals to continuously update their decision policies.

### Decision confidence modulates subsequent decision bound

Choice behavior exhibits substantial intrinsic variability (for review, see *Wyart and Koechlin, 2016*). Current models of decision-making account for this behavioral variability in terms of parameters quantifying random 'noise' in the decision process (e.g., within the DDM: drift rate variability; *Ratcliff and McKoon, 2008*). Recent evidence shows that some of this variability is not actually noise, but rather due to dynamic variations in systematic decision biases due to choice history (*Urai et al., 2017*) or arousal (*de Gee et al., 2017*). The current work extends these insights by

demonstrating that across-trial variations in decision bound are governed by decision confidence. A key factor of the current work was that observers did not receive direct feedback about their accuracy. Consequently, observers rely on an internal estimate of accuracy to generate a speed-accuracy tradeoff policy for the subsequent trial.

The model fits in *Figures 3* and *5* suggest that the effect is rather consistent across participants. For example, the increased decision bound following perceived errors in Experiments 1, 2 and 3 is found for all but one, two, and four participants, respectively. However, these model fits are realized by relying on a hierarchical Bayesian version of the DDM (*Wiecki et al., 2013*). One advantage of this method is that participants with low trial counts in specific conditions, due to the idiosyncratic nature of confidence judgments, can contribute to the analysis: Data are pooled across participants to estimate posterior parameter estimates, whereby the data of a participant with low trial counts in a specific condition will contribute less to the posterior distribution of the respective condition. Individual-subject estimates are constrained by the group posterior (assumed to be normally distributed), and estimates with low trial counts are pulled towards the group average. A limitation of this procedure is that it precludes strong conclusions about the parameter estimates from individual participants. Future studies should collect extensive data from individual participants in order to shed light on individual differences in confidence-induced bound changes.

Trial-to-trial variations in decision confidence likely result from several factors. For example, confidence might be low because of low internal evidence quality (i.e., low drift rate) or because insufficient evidence has been accumulated before committing to a choice (i.e., low decision bound). When the bound is low and results in low confidence, it is straightforward to increase the bound for the subsequent decision in order to improve performance. When drift rate is low, increasing the subsequent bound might increase accuracy only slightly, but at a vast cost in terms of response speed. Future work should aim to unravel to what extent strategic changes in decision bound differ between conditions in which variations in confidence are driven by a lack of accumulated evidence or by a lack of instantaneous evidence quality.

In all three datasets, several trials were characterized by high certainty about errors, which indeed predicted significant below-chance levels of accuracy. This observation suggests an important role for post-decisional processes, as perception of an error by definition can only occur following the commitment to a choice. Within the framework of sequential sampling models of decision-making, changes-of-mind about the perceived correct response have been explained by allowing post-decisional accumulation of evidence, coming from a sensory buffer (*Resulaj et al., 2009*) or from additional sensory input (*Fleming et al., 2018*). After the integrated evidence has hit a decision bound, and a choice is made, the evidence continues to accumulate, and so the decision variable can eventually favor the unchosen option. Such post-decisional evidence accumulation can naturally account for dissociations between confidence ratings and choice accuracy (*Moran et al., 2015*; *Navajas et al., 2016*; *Pleskac and Busemeyer, 2010*). Indeed, recent work using a similar protocol like our Experiment one showed, likewise, low confidence judgments predicting close to 0% accuracy, which was attributed to the integration of post-decisional evidence into confidence judgments (*Fleming et al., 2018*). That previous study also showed near-perfect integration of pre-decisional and post-decisional stimulus information into confidence judgments. By contrast, in our Experiment 1, we found that post-decisional sensory stimuli did not have a larger impact on confidence than a post-decisional delay with just a blank screen. The fact that the post-decisional blank and the post-decisional evidence conditions showed indistinguishable confidence judgments, indicates that post-decisional evidence was accumulated from a buffer, whereas extra sensory information was not used for the confidence judgment, different from *Fleming et al. (2018)*. This difference might be explained by a number of differences between the experimental protocols – most importantly, the fact that Fleming et al., but not us, rewarded their participants based on the accuracy of their confidence judgments, which might have motivated their participants to actively process the post-decisional stimulus information. This evidence for post-decisional contributions to confidence ratings (and in particular, certainty about errors) does not rule out the contribution of pre- and intra-decisional computations to confidence (e.g., *Gherman and Philiastides, 2018*; *Kiani and Shadlen, 2009*).

Previous work has indicated that the error positivity (Pe) tracks post-decisional evidence accumulation (*Murphy et al., 2015*) and reflects variations in decision confidence (*Boldt and Yeung, 2015*). We here demonstrated that the Pe predicted increases in subsequent decision bound. Interestingly,

this relation was specific for the Pe, and not evident for another signal reflecting confidence and error processing, the ERN. Other work has linked frontal theta oscillations, which have been proposed to drive the ERN (*Cavanagh and Frank, 2014*; *Yeung et al., 2007*; but see *Cohen and Donner, 2013*), to slowed reaction times following an error (*Cavanagh et al., 2009*). Although this is typically observed in flanker tasks, where there is no ambiguity concerning choice accuracy, a similar process of post-decision evidence accumulation has been proposed to underlie both error awareness (*Murphy et al., 2015*) and graded levels of confidence (*Pleskac and Busemeyer, 2010*). Further head-to-head comparison of participants who perform both tasks seems necessary to further resolve this discrepancy.

## The relation between decision confidence and drift rate

The main focus of the current work was to unravel influences of decision confidence on subsequent decision bound; we had no predefined hypothesis about whether confidence also affects subsequent drift rate. In Experiments 2 and 3, we observed a small reduction in drift rate following low-confidence trials. This non-monotonic reduction in drift rate driven by low confidence seems hard to reconcile with the clear monotonic relation between current-trial Pe amplitude and subsequent drift rate seen in the EEG data of Experiment 2. One explanation for this discrepancy might be that neural recordings provide a more veridical measure of the internal evaluation of accuracy than explicit confidence reports, which is subject to differences in scale use and differences in calibration. Indeed, when we fitted a model in which subsequent drift rate was allowed to vary as a function of both decision confidence and binned Pe amplitude, both the non-monotonic relation with decision confidence and the monotonic relation with Pe amplitude were replicated. Previous work has observed similar reductions of subsequent drift rate after errors (*Notebaert et al., 2009*; *Purcell and Kiani, 2016*), possibly reflecting distraction of attention from the main task due to error processing. Thus, in addition to affecting subsequent decision bounds, internal confidence (in particular: error) signals might also affect subsequent attentional focus on subsequent trials. However, given that this finding was not consistently observed across the three Experiments, in contrast with the modulation of decision bound, conclusions about the modulation of drift rate should be made with caution and warrant further investigation.

## Relation to previous work on error-dependent behavioral adjustments

Human observers slow down following incorrect choices, a phenomenon referred to as post-error slowing (*Rabbitt, 1966*). The underlying mechanism has been a matter of debate. Post-error slowing has been interpreted as a strategic increase in decision bound in order to avoid future errors (*Dutilh et al., 2012a*; *Goldfarb et al., 2012*; *Holroyd et al., 2005*) or an involuntary decrease in attentional focus (e.g., reduced drift rate) following an unexpected event (*Notebaert et al., 2009*; *Purcell and Kiani, 2016*). A key observation of the current work is that similar adjustments can also be observed based on internally computed and graded confidence signals. Our results also go beyond established effects of post-error slowing in that we establish them for trial-to-trial variations in internally computed, graded confidence signals *within* the 'correct' and 'error' condition. This aspect sets our work apart from previous model-based investigations of post-error slowing (e.g., *Dutilh et al., 2012a*; *Goldfarb et al., 2012*; *Purcell and Kiani, 2016*) and is important from an ecological perspective: internal, graded confidence signals enable the adjustment of decision parameters also in the absence of external feedback, and even after decisions that happened to be correct but were made with low confidence.

Another important novel aspect of our work is the observation of a neural confidence-encoding signal measured over parietal cortex predictive of changes in decision bound on the next trial. This observation differs from the results of a previous study into the post-error slowing in monkey lateral intraparietal area (LIP; *Purcell and Kiani, 2016*) in a critical way: *Purcell and Kiani (2016)* found that errors are followed by changes in LIP dynamics on the subsequent trial, which explained the subsequent changes in drift rate and bound; in other words, the LIP effects reported by *Purcell and Kiani (2016)* reflected the *consequences* of post-error adjustments. By contrast, the current work uncovered a putative neural *source* of adaptive adjustments of decision-making overlying parietal cortex. While the neural generators of the Pe are unknown and potentially widespread, our finding

implicates parietal cortex (along with possibly other brain regions) in the neural pathway controlling ongoing top-down adjustments of decision-making.

Modulating the speed-accuracy tradeoff by decision confidence can be thought of as an adaptive way to achieve a certain level of accuracy. Indeed, normative models prescribe that uncertainty (i.e., the inverse of confidence) should determine how much information needs to be sampled (*Bach and Dolan, 2012*; *Meyniel et al., 2015*). The current findings help bridge between studies of top-down control and perceptual decision-making (*Shea et al., 2014*; *Shimamura, 2008*; *Yeung and Summerfield, 2012*). Decision confidence has been shown to guide study choices (*Metcalfe and Finn, 2008*) and act as an internal teaching signal that supports learning (*Guggenmos et al., 2016*). Moreover, the current findings bear close resemblance to previous work showing that participants request more information when having low confidence in an impending choice (*Desender et al., 2018*). Conceptually, both the previous study and the current work demonstrate that participants sample more information when they are uncertain, which depending on the task context is achieved by increasing the decision bound or by actively requesting more information, respectively. Further evidence linking both lines of work comes from the observation that the same post-decisional neural signature of confidence, the Pe, predicts increases in decision bound (current work) and information-seeking (*Desender et al., 2019*). Interestingly, decision confidence seems to have no direct influence on top-down controlled processes such as response inhibition (*Koizumi et al., 2015*) or working memory (*Samaha et al., 2016*). Of direct relevance for the current work is a recent study by *van den Berg et al. (2016)* who showed that confidence acts as a bridge in multi-step decision-making. In their work, reward was obtained only when two choices in trial sequence were correct. The results showed a linear increase in decision bound with increasing confidence in the first decision of a sequence. The sign of this relation was opposite to what we observed in the current work. Given the multi-step nature of the task, observers likely sacrificed performance on the second choice (by decreasing the decision bound) when having low confidence in the first choice, given that both choices needed to be correct in order to obtain a reward. Contrary to this, in our current work observers were motivated to perform well on each trial, and thus adaptively varied the height of the decision bound in order to achieve optimal performance.

In sum, we have shown that decision confidence affects subsequent decision bounds on a trial-by-trial level. A post-decisional brain signal sensitive to decision confidence predicted this adaptive modulation of the decision bound at a single-trial level.

## Materials and methods

### Participants
Thirty participants (two men; age: *M* = 18.5, *SD* = 0.78, range 18–21) took part in Experiment 1 (two excluded due to a lack of data in one of the confidence judgments). Sixteen participants (eight females, age: *M* = 23.9, range 21–30) took part in Experiment 2. ERPs and non-overlapping analyses from Experiment two have been published earlier (*Boldt and Yeung, 2015*). Experiment three was a combination of two very similar datasets (see below) that are reported as one in the main text. Twelve participants (three men, mean age: 20.6 years, range 18–42) took part in Experiment 3a (one excluded due to a lack of data in one of the confidence judgments) and twelve participants (all female, mean age: 19.1 years, range 18–22) in Experiment 3b, all in return for course credit. All participants provided written informed consent before participation. All reported normal or corrected-to-normal vision and were naive with respect to the hypothesis. All procedures were approved by the local ethics committees.

### Stimuli and apparatus
In all experiments, stimuli were presented on a gray background on a 20-inch CRT monitor with a 75 Hz refresh rate, using the MATLAB toolbox Psychtoolbox3 (Brainard, 1997). Responses were made using a standard QWERTY keyboard.

In *Experiment 1*, random moving white dots were drawn in a circular aperture centered on the fixation point. The experiment was based on code provided by *Kiani et al. (2013)*, and parameter details can be found there.

In *Experiment 2*, two fields were presented with one field containing 45 dots in a 10-by-10 matrix, the other containing 55 dots. Within this constraint, the displays were randomly generated for each new trial.

In *Experiment 3*, each stimulus consisted of eight colored shapes spaced regularly around a fixation point (radius 2.8˚ visual arc). To influence trial difficulty, both color mean and color variance were manipulated. The mean color of the eight shapes was determined by the variable *C*; the variance across the eight shapes by the variable *V*. The mean color of the stimuli varied between red ([1, 0, 0]) and blue ([0, 0, 1]) along a linear path in RGB space ([*C*,0, 1 −*C*]). In Experiment 3a, *C* could take four different values: 0.425, 0.4625, 0.5375 and 0.575 (from blue to red, with 0.5 being the category boundary), and *V* could take three different values: 0.0333, 0.1000 and 0.2000 (low, medium and high variance, respectively). In Experiment 3b, *C* could take four different values: 0.450, 0.474, 0.526 and 0.550, and *V* could take two different values: 0.0333 and 0.1000. On every trial, the color of each individual element was pseudo-randomly selected with the constraint that the mean and variance of the eight elements closely matched (criterion value = 0.001) the mean of *C* and its variance *V*, respectively. Across trials, each combination of *C* and *V* values occurred equally often. The individual elements did not vary in shape.

## Procedure

### Experiment 1

After a fixation cross shown for 1000 ms, randomly moving dots were shown on the screen until a response was made or 3 s passed. On each trial, the proportion of dots moving coherently towards the left or right side of the screen was either 0%, 5%, 10%, 20% or 40%. In each block, there was an equal number of leftward and rightward movement. Participants were instructed to respond as quickly as possible, deciding whether the majority of dots were moving left or right, by pressing 'c' or 'n' with the thumbs of their left and right hand, respectively (counterbalanced between participants). When participants failed to respond within 3 s, the trial terminated with the message 'too slow, press any key to continue'. When participants responded in time, either a blank screen was shown for 1 s or continued random motion continued for 1 s (sampled from the same parameters as the pre-decisional motion). Whether a blank screen or continued motion was shown depended on the block that participants were in. Subsequently, a 6-point confidence scale appeared with labels 'certainly wrong', 'probably wrong', 'maybe wrong', 'maybe correct', 'probably correct', and 'certainly correct' (reversed order for half of the participants). Participants had unlimited time to indicate their confidence by pressing one of six numerical keys at the top of their keyboard (1, 2, 3, 8, 9 or 0), which mapped onto the six confidence levels. On half of the trials, the coherence value on each timeframe was sampled from a normal distribution (*SD* = 25.6%) around the generative coherence (cf. *Zylberberg et al., 2016*). This manipulation was irrelevant for the current purpose, however, and was ignored in the analysis. Apart from the blocks with a 1 s blank screen and 1 s continued evidence following the response, there was a third block type in which participants jointly indicated their choice (left or right) and level of confidence (low, medium, or high) in a single response. Because perceived errors cannot be indicated using this procedure, these data were omitted from all further analysis. The block order of these three conditions was counterbalanced using a Latin square. The main part of Experiment 1 comprised 9 blocks of 60 trials. The experiment started with one practice block (60 trials) without confidence judgments (only 20% and 40% coherence) that was repeated until participants reached 75% accuracy. Feedback about the accuracy of the choice was shown for 750 ms. The second practice block (60 trials) was identical to the first, except that now the full range of coherence values was used. This block was repeated until participants reached 60% accuracy. The third practice block (60 trials) was identical to the main experiment (i.e., with confidence judgments and without feedback).

### Experiment 2

On each trial, participants judged which of two simultaneously flashed fields (160 ms) contained more dots, using the same response keys as in Experiment 1 (counterbalanced across participants). After their response, a blank screen was presented for 600 ms after which confidence in the decision was queried using the same labels and response lay-out as in Experiment 1. The inter-trial interval lasted 1 s. Each participant performed 18 blocks of 48 trials. The experiment started with one

practice block with feedback without confidence judgments but with performance feedback (48 trials), and one practice block with confidence judgments but without feedback (48 trials).

## Experiment 3

This experiment was a combination of two highly similar datasets. Because both datasets show highly similar results (*Figure 5—figure supplement 5*) they were discussed as one experiment here. In both experiments, after a fixation point shown for 200 ms, the stimulus was flashed for 200 ms, followed again by the fixation point. Participants were instructed to respond as quickly as possible, deciding whether the average of the eight elements was blue or red, using the same response layout as in Experiment 1 (counterbalanced between participants). When participants failed to respond within 1500 ms, the trial terminated with the message 'too slow, press any key to continue'. When participants responded in time, a fixation point was shown for 200 ms. Then, participants where queried for a confidence judgments using the same scale and response lay-out as in Experiment 1. The inter-trial interval lasted 1000 ms. The main part of Experiment 3a comprised 8 blocks of 60 trials. To maintain a stable color criterion over the course of the experiment, each block started with 12 additional practice trials with auditory performance feedback in which the confidence judgment was omitted. The experiment started with one practice block (60 trials) without confidence judgments but with auditory performance feedback and one practice block (60 trials) with confidence judgments but without feedback. The main part of Experiment 3b comprised 8 blocks of 64 trials. Each block started with 16 additional practice trials with auditory performance feedback in which the confidence judgment was omitted. The experiment started with one practice block (64 trials) without confidence judgments but with auditory performance feedback, and one practice block (64 trials) with confidence judgments but without feedback. In even blocks of Experiment 3b, participants did not provide a confidence judgment, these data are excluded here.

## Behavioral analyses

Behavioral data were analyzed using mixed regression modeling. This method allows analyzing data at the single-trial level. We fitted random intercepts for each participant; error variance caused by between-subject differences was accounted for by adding random slopes to the model. The latter was done only when this increased the model fit, as assessed by model comparison using BIC scores. RTs and confidence were analyzed using linear mixed models, for which *F* statistics are reported and the degrees of freedom were estimated by Satterthwaite's approximation (*Kuznetsova et al., 2014*). Accuracy was analyzed using logistic linear mixed models, for which $X^2$ statistics are reported. Model fitting was done in R (R Development Core Team, 2008) using the lme4 package (*Bates et al., 2015*).

## EEG data preparation

Precise details about the EEG collection have been described in *Boldt and Yeung (2015)* and are not reiterated here. From the data presented in that work, we extracted raw-data single-trial amplitudes using the specified time windows and electrodes. Raw data were low-pass filtered at 10 Hz. Afterwards, single-trial ERN amplitudes were extracted at electrode FCz during the window −10 ms pre until 90 ms post-response. Single-trial Pe amplitudes were extracted at electrode Pz during a window from 250 ms to 350 ms post-response.

## Drift diffusion modeling

We fitted the drift diffusion model (DDM) to behavioral data (choices and reaction times). The DDM is a popular variant of sequential sampling models of two-choice tasks (*Ratcliff and McKoon, 2008*). We used the hierarchical Bayesian model fitting procedure implemented in the HDDM toolbox (*Wiecki et al., 2013*). The HDDM uses Markov-chain Monte Carlo (MCMC) sampling for generating posterior distributions over model parameters. The Bayesian MCMC generates full posterior distributions over parameter estimates, quantifying not only the most likely parameter value but also uncertainty associated with each estimate. Due to the hierarchical nature of the HDDM, estimates for individual subjects are constrained by group-level prior distributions. In practice, this results in more stable estimates for individual subjects, allowing the model to be fit even with unbalanced data, as is typically the case with confidence judgments.

For each variant of the model, we ran 10 separate Markov chains with 10000 samples each. The first half of these samples were discarded as burn-in and every second sample was discarded for thinning, reducing autocorrelation in the chains. All chains of a model were then concatenated. Group level chains were visually inspected to ensure convergence. Additionally, Gelman-Rubin R hat statistics were computed (comparing within-chain and between-chain variance) and it was checked that all group-level parameters had an R hat between 0.98–1.02. Because individual parameter estimates are constrained by group-level priors, frequentist statistics are inappropriate because data are not independent. The probability that a condition differs from another (or from the baseline) can be computed by calculating the overlap in posterior distributions. Linear relations were assessed using Friedman's χ2 test, a non-parametric rank-order test suited for repeated measures designs.

To compute statistics, we subtracted group posterior distributions of confidence on $trial_{n+2}$ from confidence on $trial_n$, and computed *p*-values from these difference distributions. To compare these models against simpler ones, we additionally fitted models in which bound, drift or both were fixed rather than free. We used Deviance Information Criterion (DIC) to compare different models to each other. Lower DIC values indicate that a model explains the data better, while taking model complexity into account. A DIC of 10 is generally taken as a meaningful difference in model fit.

## Modeling the link between confidence ratings and subsequent behavior

In Experiment 1, we first used the default accuracy coding scheme to fit a model where drift rate depended on the coherence level. All other parameters were not allowed to vary. This fit produced lower DIC values compared to a fit in which the drift rate was fixed (ΔDIC = −3882). Next, we used the regression coding scheme and allowed both the decision bound and drift rate to vary as a function of confidence on $trial_n$ and confidence on $trial_{n+2}$ (both of which were treated as factors). To obtain reliable and robust estimates, we combined trials labeled as 'certainly correct' and 'probably correct' into a 'high confidence' bin, trials labeled as 'guess correct' and 'guess wrong' into a 'low confidence' bin, and trials labeled as 'probably wrong' and 'certainly wrong' into a 'perceived error' bin. This ensured a sufficient number of trials for each level of confidence for all individual participants (high confidence: *M* = 191.7, range 47–311; low confidence: *M* = 114.5, range 4–228; perceived error: *M* = 23.8, range 1–83). The hierarchical Bayesian approach does not fit the model to individual subject's data, but rather it jointly fits the data of the entire group. Therefore, data from participants with low trial counts in certain conditions does not contribute much to the posteriors for the respective condition. At the same time, participant-level estimates are estimated, but these are constrained by the group-level estimate. One obvious advantage of this approach is that participants with unequal trial numbers across conditions can contribute to the analysis, whereas in traditional approaches their data would be lost.

Trials with high confidence were always treated as reference category (i.e., fixed to zero). In addition, the drift rate was allowed to vary as a function of coherence, which was treated as a covariate (because we were not interested in the parameter estimate but solely wanted to capture variance in the data accounted for by signal-to-noise ratio). To quantify the influence of confidence on the subsequent decision bound and drift rate, we subtracted estimates of subsequent bound and drift by confidence on $trial_{n+2}$ from estimates of subsequent bound and drift by confidence on $trial_n$. Statistics of the simple effects of confidence on $trial_n$ and confidence on $trial_{n+2}$ are reported in the figure supplements. Relative to the null model without confidence, the full model (presented in *Figure 3*) provides the best fit (ΔDIC = −288), explaining the data better than simpler models in which only the bound (ΔDIC = −234) or the drift (ΔDIC = −94) were allowed to vary.

The data of Experiment two were analyzed in the same way, except that difficulty was fixed and thus trial difficulty (i.e., coherence or signal-to-noise ratio) needed not to be accounted for within the model. Relative to the null model (presented in *Figure 5*), allowing both drift and bound to vary as a function of confidence provides the best fit (ΔDIC = −677), which explained the data better than simpler models in which only the bound (ΔDIC = −302) or the drift (ΔDIC = −170) were allowed to vary. Trial counts for this experiment were relatively high (high confidence: *M* = 516, range 132–705; low confidence: *M* = 257, range 77–674; perceived error: *M* = 80, range 18–197).

The data of Experiment three were analyzed in the same way as Experiment 1, except that the variable coherence was replaced by signal-to-noise ratio. For both experiment 3a and 3b, a model in which only the drift was allowed to vary as a function of signal-to-noise ratio produced lower DIC values compared to a fit in which the drift rate was fixed (Experiment 3a: ΔDIC = −311; Experiment 3b:

ΔDIC = −88). For the confidence-dependent fitting, a single model was fit to the data of Experiments 3a and 3b simultaneously. Relative to the null model without confidence, the full model (presented in *Figure 5*) provides the best fit (ΔDIC = −634), explaining the data better than simpler models in which only the bound (ΔDIC = −58) or the drift (ΔDIC = −260) were allowed to vary. Trial counts were relatively high (high confidence: $M$ = 166, range 92–302; low confidence: $M$ = 92, range 41–175; perceived error: $M$ = 38, range 10–41).

## Modeling the link between ERP components and subsequent behavior

Because single-trial EEG contains substantial noise, a robust measure was computed by rank ordering all trials per participant, and then using rank as a predictor rather than the raw EEG signal. A hierarchical DDM regression model was then fit in which subsequent bound and drift were allowed to vary as a function of the Pe and the ERN, both on $trial_n$ and $trial_{n+2}$.

To examine potential nonlinear effects, the Pe was divided into five bins, separately for each participant. This was done after regressing out the effect of the ERN, separately for each participant. Then, a hierarchical drift diffusion model regression model was run in which subsequent bound and drift were allowed to vary as a function of binned Pe on $trial_n$ and $trial_{n+2}$. The bin with the lowest amplitudes was always treated as the reference category. Model comparison revealed that, relative to a model without the Pe, the full model provides the best fit (ΔDIC = −6524), explaining the data much better than simpler models in which only the bound (ΔDIC = −1816) or only the drift (ΔDIC = −1787) were allowed to vary as a function of Pe. When applying the same binned analysis to the ERN, model comparison revealed that both the full model (ΔDIC = 24), and a model in which only drift (ΔDIC = 171) or bound (ΔDIC = 182) were allowed to vary, provided a worse fit than the null model. Thus, the ERN had no explanatory power in explaining either drift rate or decision bound.

## Controlling for autocorrelation in performance

The relation between decision confidence and decision bound on the subsequent trial might be confounded by autocorrelations in performance. During the course of an experiment autocorrelation is typically observed in RTs, accuracy (*Dutilh et al., 2012b*; *Gilden, 2003*; *Palva et al., 2013*), and confidence (*Rahnev et al., 2015*). This could be due to slow drifts in behavioral state factors (e.g. motivation, arousal, attention). When observers report high confidence in 'fast periods' and low confidence in 'slow periods' of the experiment (c.f., the link between response speed and confidence; *Kiani et al., 2014*), this can artificially induce a negative relation between decision confidence on $trial_n$ and reaction time on $trial_{n+1}$. We reasoned that one solution to control for such effects of slow drift is to is use confidence on $trial_{n+2}$: Confidence on $trial_{n+2}$ cannot causally affect decision bound on $trial_{n+1}$ (because of temporal sequence) and might thus be used as a proxy of the effects of slow performance drifts. So, we isolated the impact of rapid (trial-by-trial) and (as we hypothesize) causally-mediated, confidence-dependent changes in decision bound from slow performance drifts as follows: we took the *difference* between confidence-dependent changes in decision bound, whereby confidence was either evaluated on $trial_n$ or confidence was evaluated on $trial_{n+2}$; we subtracted the latter (proxy of drift) from the former.

A possible concern is that the decision bound on $trial_{n+1}$ affected confidence ratings on $trial_{n+2}$, which would complicate the interpretation of the results of our approach. Thus, we also used a complementary approach controlling for slow drifts in performance, which is analogous to an approach established in the post-error slowing literature (*Dutilh et al., 2012b*; *Purcell and Kiani, 2016*). In that approach, post-error trials are compared to post-correct trials that are also pre-error. As a consequence, both trial types appear adjacent to an error, and therefore likely stem from the same time period in the Experiment. We adopted this approach to confidence ratings as follows: decision bound and drift rate on $trial_{n+1}$ were fitted in separate models where i) we compared the effect of low confidence on $trial_n$ to high confidence on $trial_n$ for which $trial_{n+2}$ was a low confidence trial, and ii) we compared the effect of perceived errors on $trial_n$ to high confidence trials on $trial_n$ for which $trial_{n+2}$ was a perceived error. Thus, this ensured that the two trial types that were compared to each other stemmed from statistically similar environments. For the EEG data, we fitted a new model estimating decision bound and drift rate on $trial_{n+1}$ when $trial_n$ stemmed from the lowest Pe amplitude quantile, compared to when $trial_n$ stemmed from the highest Pe amplitude quantile *and* $trial_{n+2}$ stemmed from the lowest Pe amplitude quantile.

## Acknowledgements

Thanks to Jan-Willem de Gee for help with the drift diffusion modeling and to Peter Murphy, Konstantinos Tsetsos, Niklas Wilming, Anne Urai and Cristian Buc Calderon for useful comments on an earlier version of the manuscript.

## Additional information

### Competing interests

Tobias H Donner: Reviewing editor, *eLife*. The other authors declare that no competing interests exist.

### Funding

| Funder | Grant reference number | Author |
| --- | --- | --- |
| Fonds Wetenschappelijk Onderzoek | FWO [PEGASUS]$^2$ Marie Skłodowska-Curie fellow | Kobe Desender |
| Economic and Social Research Council | PhD studentship | Annika Boldt |
| Wellcome | Sir Henry Wellcome Postdoctoral Fellowship | Annika Boldt |
| Deutsche Forschungsgemeinschaft | DO 1240/2-1 | Tobias H Donner |
| Deutsche Forschungsgemeinschaft | DO 1240/3-1 | Tobias H Donner |
| Fonds Wetenschappelijk Onderzoek | G010419N | Kobe Desender Tom Verguts |
| Deutsche Forschungsgemeinschaft | SFB 936/A7 | Tobias H Donner |

The funders had no role in study design, data collection and interpretation, or the decision to submit the work for publication.

### Author contributions

Kobe Desender, Conceptualization, Data curation, Formal analysis, Investigation, Visualization, Writing—original draft; Annika Boldt, Resources, Data curation, Writing—review and editing; Tom Verguts, Tobias H Donner, Conceptualization, Resources, Supervision, Writing—review and editing

### Author ORCIDs

Kobe Desender (ID) https://orcid.org/0000-0002-5462-4260
Tobias H Donner (ID) http://orcid.org/0000-0002-7559-6019

### Ethics

Human subjects: Written informed consent and consent to publish was obtained prior to participation. All procedures were approved by the local ethics committee of the University Medical Center, Hamburg-Eppendorf (PV5512).

### Decision letter and Author response

Decision letter https://doi.org/10.7554/eLife.43499.030
Author response https://doi.org/10.7554/eLife.43499.031

## Additional files

### Supplementary files
• Transparent reporting form
DOI: https://doi.org/10.7554/eLife.43499.026

### Data availability
All data have been deposited online and can be freely accessed (https://osf.io/83x7c/ and https://github.com/AnnikaBoldt/Boldt_Yeung_2015). All analysis code is available online (https://github.com/donnerlab/2019_eLife_Desender).

The following dataset was generated:

| Author(s) | Year | Dataset title | Dataset URL | Database and Identifier |
|---|---|---|---|---|
| Desender K, Boldt A, Verguts T, Donner TH | 2019 | Dataset: Post-decisional sense of confidence shapes speed-accuracy tradeoff for subsequent choices | https://osf.io/83x7c/ | Open Science Framework, 83x7c |

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
