## [Decision Letter]

Thank you for submitting your article "Post-decisional sense of confidence shapes speed-accuracy tradeoff for subsequent choices" for consideration by *eLife*. Your article has been reviewed by three peer reviewers, and the evaluation has been overseen by a Reviewing Editor and Michael Frank as the Senior Editor. The following individual involved in review of your submission has agreed to reveal their identity: Simon P. Kelly (Reviewer #2).

The reviewers have discussed the reviews with one another and the Reviewing Editor has drafted this decision to help you prepare a revised submission.

Summary:

This paper tackles an interesting topic concerning how trial-to-trial adjustments to decision policy are made based on confidence. These policy adjustments are well studied for categorical (error/correct) outcomes of past trials but less understood for graded levels of reported confidence for each outcome. The authors have compiled a substantial dataset to study changes of decision bound with confidence associated with past decisions. Their dataset consists of three behavioral experiments that used different perceptual decision tasks. One experiment also included EEG data. They report that perceived error trials are associated with a higher subsequent decision bound than high confidence trials. They also report a strong, linear association between the Pe ERP component and subsequent decision bound. The paper is quite interesting and written well. However, as explained below, there are substantial conceptual and technical complexities that should be addressed before publication.

Essential revisions:

1) What supports the claim that it is only the "post-decisional" component of confidence (as opposed to the entirety of confidence) that influences subsequent choices? As the authors show in Figure 2 and supplementary figures, confidence of the current choice is influenced by processing the evidence both before and after the choice is reported. However, the pre-decision component and its potential effects are largely ignored. The authors seem to rely on the late EEG correlate for their claim, but that point is a bit unclear from the manuscript. Also, a late correlate does not necessarily imply only post-decisional contributions. Because the title and Abstract of the manuscript emphasize post-decisional effects, this claim should be properly supported through re-analysis of existing EEG and behavioral data (or a new behavioral experiment, if needed). Alternatively, the authors could focus on the effects of confidence as a whole. The reviewers find the paper interesting enough even if the post-decisional claims are dropped.

2) The claim that speed-accuracy tradeoff is altered by confidence needs clarification and support. As shown in Figures 3 and 5, changes of accuracy are quite small (<5%) and it is unclear if they reach statistical significance. Similarly, changes of RT are quite small and their statistical significance remains unclear. Because the DDM is only an approximation to the neural computations that underlie the behavior, it is critical that claims about speed-accuracy tradeoff are supported by changes in speed and/or accuracy, instead of only changes in model parameters. Also, the data suggest that behavioral effects of confidence on subsequent decisions may be more complex than a simple trade off of speed and accuracy. For example, it seems that effects of confidence on subsequent decision bound were stronger for error trials than for correct trials. Is that right, and if so what is the interpretation?

3) It is critical to show that apparent effects of confidence on behavior are not mediated by other factors. The reviewers wonder whether there is something that varies from trial to trial, such as coherence, coherence volatility, or dot-clustering which could both affect confidence on the current trial and also prime the subject to adjust on the next trial, without a direct impact of confidence on the adjustment. Take volatility in experiment 1 for example: On half of trials coherence varied randomly from frame to frame, which would increase variance in the decision variable and thus plausibly affect confidence and also the setting of bound on the next trial. In the Materials and methods the authors simply state that this volatility manipulation doesn't matter, but the main relationship between confidence and bound on next trial could be mediated through this variation. Coherence itself could also be a driver of the relationship – to the extent that subjects can tell high from low coherence trials, they may lower their bound after low coherence trials and increase it after high coherence trials – this might not be beneficial but neither are many typical priming effects. Taking the error trials for example, trials in which the subjects most thought they were correct would be the low coherence trials, which on the basis of coherence itself induce a bound lowering, not necessarily mediated through confidence. Support for this interpretation comes from experiment 2, where stimulus difficulty was fixed. In this experiment, there was not a robust difference between high and low confidence decision bounds in Figures 5C or 6A/B. To support their conclusion, the authors should carefully consider mediating factors. They may be able to conduct mediation analyses to better test for confidence's direct impact despite such other factors, and may be able to explore the factors themselves to see if they have any such impact in the first place, e.g. testing for differences across coherence levels, volatility levels, and even within-condition motion energy or dot distribution.

4) The composition of the dataset raises concerns. Three issues require clarification:

a) Did participants receive adequate practice prior to data collection to learn the tasks and show stable performance? In experiment 1, data collection started following only 180 practice trials. It is unclear that the practice included feedback and was sufficient for subjects to learn how to optimally use the sensory evidence or how to generate reliable confidence reports. Was the behavior stable throughout data collection? More important, how good were the data? Reaction times of a relatively large fraction of subjects seem to vary minimally across stimulus strengths in Figure 2. Also, many subjects seem to have accuracies far below 100% for the strongest stimuli in the experiment. Figure 2D suggests that ~20% of subjects have almost chance level accuracy even when they report they are quite sure to be correct! Similar problems – shortage of practice trials and unclear quality and stability of performance – are present in the other experiments too.

b) What fraction of subjects support the main results? The analysis methods of the paper prevent a clear answer to this question. Consequently, we cannot tell that the average trends in the population are not generated by a mixture of diverse (or even opposing) trends across participants. For example, Figures 3 or 5 could include subjects that go against the population average for changes of RT and accuracy, or changes of decision bound. Can the authors clarify if individual participant's results match the population average and whether there are subjects that deviate from the average trends?

c) Can the authors clarify the trial numbers in subsection “Decision confidence influences subsequent decision bound”? Does 83-1 mean between 1 and 83 trials? If yes, how could 1 trial be sufficient for fitting the model to individual subject's data?

5) The reviewers appreciate the authors' attempt to remove confounds caused by slow fluctuations of RT, accuracy and confidence. However, it is unclear that the correction procedure is adequate. Three related concerns are raised:

a) It is unclear that the underlying assumption of the analysis is correct. Of course confidence on trial n+2 cannot influence the decision bound on n+1, but could the decision bound on n+1 influence confidence on n+2? In both directions, these are just correlations, What support is there for the causal claim the authors are making of confidence on trial n changing bound on n+1?

b) The motivation of the analysis is to remove generic autocorrelations from the data (due to, e.g., long periods of high/low confidence correlated with low/high RT), under the assumption that trial n+2 confidence cannot be causally associated with trial n+1 decision bound. However, based on the unnormalized, "simple effects of confidence on decision bound," the effects seem to be mainly coming from trial n+2, rather than trial n (e.g., Figure 5—figure supplement 1, Figure 8—figure supplement 4, and Figure 5—figure supplement 3). This is fairly consistent in the behavioral data and is especially striking for the relation between Pe amplitude and decision bound. How does it affect whether we believe trial n+2 is an appropriate baseline measure? More important, these observations appear to challenge the main claim that confidence-dependent bound changes are shaped by trial n. Can the authors clarify why this challenge is not critical?

c) The authors use the same model to quantify the effect of trial n or n+2 on trial n+1 and subtract the two effects to filter out slow fluctuations. This method would be successful only to the extent that the applied models are good at fitting the data. If a model fails to adequately capture the data, the residual error could contain slow fluctuations of behavior asymmetrically for the n/n+1 and n+2/n+1 analyses, which could contribute to results. There are reasons to be concerned because we do not know how well the DDM fits the behavior of individual participants, especially their RT distributions (not just the mean RT across the population; also consider large lapses and shallow RT functions). Showing residual errors of the model for RT and the autocorrelation of these residual errors would be useful to alleviate this concern.

6) A point that deserves discussion and possibly looking into is the role of RT in determining the relationship between confidence and subsequent adjustment. From past papers, in varying-difficulty contexts, lower confidence can result from a decision reaching commitment at too low a level of cumulative evidence, OR alternatively from a crossing made later in the trial. The latter is more an indication that the current trial had weaker evidence than that the bound was not high enough. Setting the bound higher for the latter kind of trial would bring an increase in accuracy but at a vast cost in terms of time. Bound increases for low confidence do not clearly seem to be universally beneficial, and it would seem to depend on the cost of time, presence of deadlines etc. This kind of territory may be covered in papers like Meyniel et al., but either way, it seems a matter worth discussing in the current paper.

---

## [Author Response]

Essential revisions:1) What supports the claim that it is only the "post-decisional" component of confidence (as opposed to the entirety of confidence) that influences subsequent choices? As the authors show in Figure 2 and supplementary figures, confidence of the current choice is influenced by processing the evidence both before and after the choice is reported. However, the pre-decision component and its potential effects are largely ignored. The authors seem to rely on the late EEG correlate for their claim, but that point is a bit unclear from the manuscript. Also, a late correlate does not necessarily imply only post-decisional contributions. Because the title and Abstract of the manuscript emphasize post-decisional effects, this claim should be properly supported through re-analysis of existing EEG and behavioral data (or a new behavioral experiment, if needed). Alternatively, the authors could focus on the effects of confidence as a whole. The reviewers find the paper interesting enough even if the post-decisional claims are dropped.

We agree that our previous focus on the post-decisional component of confidence was overly specific and not necessary for our main conclusion. We have dropped the corresponding claims from the title, Abstract, and Results section and keep this as a point for the Discussion section.

Our focus was motivated by the observation that perceived errors played a critical role in the bound adjustments. Assuming that subjects base both their choices and confidence ratings on (some transformation of) the same decision variable, perceived errors are very likely to result from post-decisional processing. However, we acknowledge that this does not rule out an influence of pre- and intra-decisional computations on confidence ratings and subsequent bound adjustments. We now acknowledge this point explicitly in the Discussion:

“This evidence for post-decisional contributions to confidence ratings (and in particular, certainty about errors) does not rule out the contribution of pre- and intra-decisional computations to confidence (e.g., Gherman and Philiastides, 2018; Kiani and Shadlen, 2009).”

2) The claim that speed-accuracy tradeoff is altered by confidence needs clarification and support. As shown in Figures 3 and 5, changes of accuracy are quite small (<5%) and it is unclear if they reach statistical significance. Similarly, changes of RT are quite small and their statistical significance remains unclear. Because the DDM is only an approximation to the neural computations that underlie the behavior, it is critical that claims about speed-accuracy tradeoff are supported by changes in speed and/or accuracy, instead of only changes in model parameters.

The effects of confidence ratings on subsequent RTs and accuracies are qualitatively in line with a change in speed-accuracy tradeoff (i.e., increase in response caution after perceived errors) for all data sets. This is statistically significant for Experiments 2 (RTs: *F*(2,45) = 4.96, *p* =.011; accuracy: *p* =.362) and Experiment 3 (RTs: *F*(2,65.99) = 3.80, *p* =.027; accuracy: *F*(2,66) = 8.09, *p* <.001), but not quite for Experiment 1 (RTs: *p* =.10; accuracy: *p* =.357), which showed overall subtler effects.

We devised an aggregate, model-free behavioral measure of speed accuracy trade-off by multiplying median RT and mean accuracy at trial_n+1_, for each level of confidence at trial_n_, Increase in bound separation predicts larger values on this model-free measure of response caution. This is conceptually similar to the so-called “inverse efficiency score” (RT/accuracy) which is often used as model-free index of drift rate (Bruyer and Brysbaert, 2011). This measure is now also presented, along with RT and accuracy separately, in Figure 3 and Figure 5, alongside the model predictions (green crosses). This measure shows a statistically significant effect of confidence ratings on subsequent speed-accuracy tradeoff for all three experiments (Experiment 1: *F*(2,81) = 3.13, *p* =.049; Experiment 2: *F*(2,45) = 3.21, *p* =.050; Experiment 3: *F*(2,66) = 14.43, *p* <.001). This is explained and reported in the Results section:

“A bound increase will increase both RT and accuracy, a trend that was evident in the data (Figure 3A, left). We multiplied median RT and mean accuracy as a function of confidence to combine both effects into a single, model-free measure of response caution (Figure 3A, right). This aggregate measure of response caution was predicted by the confidence rating from the previous decision, F(2,81) = 3.13, p =.049. Post-hoc contrasts showed increased caution after perceived errors compared to after both high confidence, z = 2.27, p =.032, and low confidence, z = 2.05, p =.041. There was no difference in caution following high and low confidence ratings, p =.823. The confidence-dependent change in subsequent response caution was explained by a DDM, in which decision bounds and drift rate could vary as a function of previous confidence (Figure 3A, green crosses; see Materials and methods).”

“As in Experiment 1, our model-free measure of response caution (RT*accuracy) was modulated by confidence ratings on the previous trial, *F*(2,45) = 3.21, *p* =.050 (perceived errors vs. high confidence: *z* = 2.53, *p* =.011; no significant differences for other comparisons *p*s >.178; Figure 5B).”

“Our model-free measure of response caution was again affected by previous confidence ratings, *F*(2,66) = 14.43, *p* <.001 (perceived errors vs. high confidence: *z* = 4.61, *p* <.001; perceived errors vs. low confidence: *z* = 4.69, *p* <.001; high vs. low confidence: *p* =.938; Figure 5B).”

Also, the data suggest that behavioral effects of confidence on subsequent decisions may be more complex than a simple trade off of speed and accuracy. For example, it seems that effects of confidence on subsequent decision bound were stronger for error trials than for correct trials. Is that right, and if so what is the interpretation?

We agree with the reviewers that it appears as if the effect of confidence on subsequent decision bound is stronger for error trials than for correct trials. However, note that the main issue here is that the effect holds when current-trial accuracy is held constant (i.e., either being correct or incorrect), so as to rule out that the increase in decision bound actually reflects post-error slowing. Because of limited number of trials when separately fitting corrects and errors, the uncertainty in each of these estimates is much larger, as visible in the width of the posteriors. This in turn has a large influence on the estimates of individual participants, which are therefore much more constrained by the group posterior and hence much more homogenous (particularly the case for current error trials).

It is not straightforward to directly compare the estimates of current correct and current error to each other, because both come from a different model. Within a single model fit, parameters can be directly compared to each other by pairwise comparison of the traces from the Monte-Carlo Markov-Chain, but comparing different models to each other cannot be done because these models generate different chains. We agree that it is a very interesting question whether confidence is associated with adjustments of decision computation more complex than changes in speed-accuracy tradeoff (and relatedly: bound separation and drift). But we feel that answering this question will require different modeling approaches (e.g. more biophysically inspired models) that are beyond the scope of the current work.

3) It is critical to show that apparent effects of confidence on behavior are not mediated by other factors. The reviewers wonder whether there is something that varies from trial to trial, such as coherence, coherence volatility, or dot-clustering which could both affect confidence on the current trial and also prime the subject to adjust on the next trial, without a direct impact of confidence on the adjustment. Take volatility in experiment 1 for example: On half of trials coherence varied randomly from frame to frame, which would increase variance in the decision variable and thus plausibly affect confidence and also the setting of bound on the next trial. In the Materials and methods the authors simply state that this volatility manipulation doesn't matter, but the main relationship between confidence and bound on next trial could be mediated through this variation. Coherence itself could also be a driver of the relationship – to the extent that subjects can tell high from low coherence trials, they may lower their bound after low coherence trials and increase it after high coherence trials – this might not be beneficial but neither are many typical priming effects. Taking the error trials for example, trials in which the subjects most thought they were correct would be the low coherence trials, which on the basis of coherence itself induce a bound lowering, not necessarily mediated through confidence. Support for this interpretation comes from experiment 2, where stimulus difficulty was fixed. In this experiment, there was not a robust difference between high and low confidence decision bounds in Figures 5C or 6A/B. To support their conclusion, the authors should carefully consider mediating factors. They may be able to conduct mediation analyses to better test for confidence's direct impact despite such other factors, and may be able to explore the factors themselves to see if they have any such impact in the first place, e.g. testing for differences across coherence levels, volatility levels, and even within-condition motion energy or dot distribution.

Thank you for raising this important point. We ran additional analyses to test whether the experimental variables that we manipulated from trial to trial affected subsequent decision bound. For Experiment 1, we fitted a model that quantified the effect of current evidence strength (coherence) and current evidence volatility on subsequent decision bound, while allowing for subsequent drift rate to depend on subsequent coherence. This analysis did not show an effect of current volatility, *p* =.242, or current coherence, *p* =.133. For each of these, zero (i.e., no effect) was included in the 95% highest density interval of the posterior distribution (-.167 to.026, and -.020 to.039, respectively), suggesting it is likely that the actual parameter value is close to zero.

Likewise, we fitted a model to the data of Experiment 3 that tested the influence of current evidence strength on subsequent decision bound, while allowing for subsequent drift rate to depend on subsequent signal-to-noise ratio. Again, there was no effect of current signal-to-noise ratio, *p* =.220, and zero was included in the 95% highest density interval of the posterior (-.010 to.031). Given that neither of these variables affected subsequent decision bound, it seems unlikely that the effect of confidence on subsequent bound is mediated by one of these factors.

These additional control analyses are now reported in the corresponding places of the Results section:

“The effects of confidence ratings on subsequent decision bounds are unlikely to be caused by our systematic manipulation of evidence strength (i.e., motion coherence) or evidence volatility (Materials and methods). Confidence ratings were reliably affected by both evidence strength (Figure 2C) and evidence volatility (data not shown, *F*(1, 26.7) = 47.10, *p* <.001). However, evidence strength and volatility, in turn, did not affect subsequent decision bound, both *p*s >.133. For each of these, zero (i.e., no effect) was included in the 95% highest density interval of the posterior distribution (-.167 to.026, and -.020 to.039, respectively), suggesting it is likely that the true parameter value was close to zero.”

“As in Experiment 1, the systematic trial-to-trial variations of evidence strength (SNR) did not influence subsequent decision bound in Experiment 3 (*p* =.220), and zero was included in the 95% highest density interval of the posterior (-.010 to. 031).”

4) The composition of the dataset raises concerns. Three issues require clarification:a) Did participants receive adequate practice prior to data collection to learn the tasks and show stable performance? In experiment 1, data collection started following only 180 practice trials. It is unclear that the practice included feedback and was sufficient for subjects to learn how to optimally use the sensory evidence or how to generate reliable confidence reports. Was the behavior stable throughout data collection? More important, how good were the data? Reaction times of a relatively large fraction of subjects seem to vary minimally across stimulus strengths in Figure 2. Also, many subjects seem to have accuracies far below 100% for the strongest stimuli in the experiment. Figure 2D suggests that ~20% of subjects have almost chance level accuracy even when they report they are quite sure to be correct! Similar problems – shortage of practice trials and unclear quality and stability of performance – are present in the other experiments too.

We agree with the reviewers that the Materials and methods should be expanded to fully unpack the procedures, and we apologize for failing to do so in our initial submission.

Several points were raised by the reviewers: a) nature of task practice, b) stability of performance, and c) the relation between confidence and accuracy (for Experiment 1 only). In what follows, we address each of these points, separately for each Experiment.

Experiment 1:

We now further clarify the details of the training blocks in the Materials and methods: “The experiment started with one practice block (60 trials) without confidence judgments (only 20% and 40% coherence) that was repeated until participants reached 75% accuracy. Feedback about the accuracy of the choice was shown for 750 ms. The second practice block (60 trials) was identical to the first, except that now the full range of coherence values was used. This block was repeated until participants reached 60% accuracy. The third practice block (60 trials) was identical to the main experiment (i.e., with confidence judgments and without feedback).”

To shed light on the quality of the data, we performed additional linear regression analyses on the raw data of each participant. These showed that coherence significantly affected RTs (*p* <.05) for 27 out of 28 participants (with a negative slope for all 28), whereas coherence significantly affected both accuracy and confidence (*p* <.05) in the expected direction for all 28 participants. Next, we added the variable experiment half (first half vs. second half) to these regressions, and tested for an interaction between coherence and experiment half. None of the 28 participants showed an interaction between experiment half and coherence in predicting accuracy, whereas this was the case for 5 participants when predicting RTs (one participant became more sensitive to coherence in the second half) and for 6 participants when predicting confidence (one participant became more sensitive to coherence in the second half). Finally, in Author response image 1 we included raw RTs and accuracy per block for the first six participants of Experiment 1, to demonstrate the stability of performance over the course of the experiment (note that this was highly similar for the other participants).

As becomes clear from Author response image 1, the data of the majority of participants was of high quality, and rather stable throughout the entire experiment. In order to keep the manuscript concise and to the point, we decided not to include these additional quality checks in the manuscript, but we are glad to do so if the reviewers deem this to be helpful. Finally, to fully satisfy the concern raised about data quality, we reran the main analysis on the subset of 17 participants who showed significant scaling of RTs, accuracy and confidence with coherence, and who did not show a significant interaction between coherence and experiment half on any of these measures. This analysis showed that subsequent decision bound separation was numerically higher when participants had low confidence in their choice (*M* =.04, *p* =.156) and significantly increased when participants perceived an error (M =.30, *p* <.001); the latter two were also different from each other (p <.001). There were no significant effects on drift rate, *p*s >.308.

Also, many subjects seem to have accuracies far below 100% for the strongest stimuli in the experiment.

Note that perfect accuracy is not to be expected here because participants were explicitly motivated to perform the experiment as fast and accurate as possible. Therefore, errors are expected even at the highest coherence level, probably because participants put their decision bound too low. Evidence for this comes from the observation that, on average, 65% of the errors committed on trials with 40% coherence (26 out of 51) were judged as *perceived errors*, suggesting that these errors do not result from poor data quality, but rather reflect genuine premature responses.

Figure 2D suggests that ~20% of subjects have almost chance level accuracy even when they report they are quite sure to be correct!

Thanks for spotting this. This figure was based on a subset of blocks in which participants jointly indicated their level of confidence and choice. There were four participants who misunderstood the instructions (i.e., they used the response scale as if it would range from certainly correct to certainly wrong, rather than ranging from certainly left to certainly right), and consequently performed at chance level. While these participants were included in the main analysis (i.e., where they did use the scale correctly and performed well in the separated choice-confidence blocks), they should obviously not have been included in these analyses. Note that reviewer 2 (comment 5) commented that these data are not a fair comparison because only half the range of the confidence scale is available (compared to the whole confidence scale in the separated choice and confidence condition), and so we decided to drop these data, given that they were not critical for our conclusion.

*Experiment 2*:

We now further clarify the details of the training blocks: “The experiment started with one practice block with feedback without confidence judgments but with performance feedback (48 trials), and one practice block with confidence judgments but without feedback (48 trials)”.

Because only a single level of difficulty was used throughout the entire experiment, the quality checks reported above cannot be carried out on these data.

Experiment 3:

We now further clarify the details of the training blocks in the Materials and Methods. To shed light on the quality of the data, the same linear regression analyses were run as for Experiment 1. These showed that signal-to-noise ratio significantly affected RTs (*p* <.05) for 7 out of 23 participants (with the expected negative slope for 19 out of 23 participants), and it significantly affected accuracy for 21 out of 23 participants (22 in the expected direction), and confidence for 18 out of 23 participants (20 in the expected direction). Note that this low number of significant effects of signal-to-noise on RTs does not necessarily imply low signal quality; because of the limited number of trials per participants it could well be that the effect is present (i.e., the slopes are in the correct direction for most participants) but experimental power is simply too low to detect these at the individual participant level. Next, as before, we added experiment half to these regressions. None of the 28 participants showed an interaction between experiment half and coherence in predicting RTs or in predicting confidence, whereas this was the case for one participant when predicting accuracy (s/he became more sensitive to SNR in the second half). Nevertheless, to fully satisfy the concern raised about data quality, we reran the main analysis on the subset of 7 participants who showed significant scaling of RTs, accuracy and confidence with coherence, and who did not show a significant interaction between coherence and experiment halve on any of these measures. This analysis showed that subsequent decision bound separation was decreased when participants had low confidence in their choice (*M* = -.08, *p* =.041) and non-significantly increased when participants perceived an error (*M* =.07, *p* =.112); the latter two were different from each other (*p* =.010). There were no significant effects on drift rate, *p*s >.124.

b) What fraction of subjects support the main results? The analysis methods of the paper prevent a clear answer to this question. Consequently, we cannot tell that the average trends in the population are not generated by a mixture of diverse (or even opposing) trends across participants. For example, Figures 3 or 5 could include subjects that go against the population average for changes of RT and accuracy, or changes of decision bound. Can the authors clarify if individual participant's results match the population average and whether there are subjects that deviate from the average trends?

Unfortunately, this question is difficult to answer given the approach we have opted to pursue for this particular paper. Upfront, we would like to clarify that there are two possible approaches in behavioral modeling studies: i) gathering a lot of data from a relatively small (N<=10) sample of participants and fit each at the individual level, using independent fits; or ii) gather fewer trials per participant, from a larger sample of participants, and then resort to hierarchical Bayesian model fitting approaches, which pool trials across participants to estimate parameters at the group level. We feel that there is no generally “right” or “wrong” approach, but the optimal choice of approach depends on the specific question at hand. Indeed, in our previous work, we have used both of these approaches.

In the present study, we opted for the second approach, because we expected substantial inter-individual differences, both in the way the confidence ratings are used and in how they are translated into adjustments of subsequent decision processing.

Confidence ratings are often unevenly distributed. Although the Bayesian fitting method is very powerful – especially when trial counts are low and/or unevenly distributed – it comes at the costs that parameters are not readily interpretable at the participant-level, because individual estimates are constrained by the group level prior.

We now acknowledge this limitation explicitly in the Discussion:

“The model fits in Figure 3 and 5 suggest that the effect is rather consistent across participants. For example, the increased decision bound following perceived errors in Experiments 1, 2 and 3 is found for all but one, two, and four participants, respectively. However, these model fits are realized by relying on a hierarchical Bayesian version of the DDM (Wiecki, Sofer, and Frank, 2013a). One advantage of this method is that participants with low trial counts in specific conditions due to the idiosyncratic nature of confidence judgments, can contribute to the analysis: Data are pooled across participants to estimate posterior parameter estimates, whereby the data of a participant with low trial counts in a specific condition will contribute less to the posterior distribution of the respective condition. Individual-subject estimates are constrained by the group posterior (assumed to be normally distributed), and estimates with low trial counts are pulled towards the group average. A limitation of this procedure is that it precludes strong conclusions about the parameter estimates from individual participants. Future studies should collect extensive data from individual participants in order to shed light on individual differences in confidence-induced bound changes.”

c) Can the authors clarify the trial numbers in subsection “Decision confidence influences subsequent decision bound”? Does 83-1 mean between 1 and 83 trials?

Yes, that is indeed the case, and it is a consequence of the approach we pursued here (see previous point). The combination of relatively few trials per participant and an uneven distribution of individual confidence ratings could lead, in few cases, to very low trial counts for certain subjects in certain confidence conditions. The hierarchical Bayesian model fitting procedure is specifically designed for situations like this (Wiecki et al., 2012).

Histograms of the trial counts for the three confidence bins are shown in Author response images 2, 3 and 4 (note, blue = high confidence, red = low confidence, green = perceived errors). Overall, trial counts smaller than 10 per subject were rare in these data sets (note that Experiment 3 comprised Exp 3a and 3b, see Methods and Materials, causing the difference in overall trial counts seen).

**Author response image 2. respfig2:** 

**Author response image 3. respfig3:** 

**Author response image 4. respfig4:** 

If yes, how could 1 trial be sufficient for fitting the model to individual subject's data?

The hierarchical Bayesian approach does not fit the model to individual subject’s data, but rather it jointly fits the data of the entire group. Therefore, data from participants with low trial counts in a certain condition does not contribute much to the posteriors for the respective condition. Participant-level estimates are estimated, but these are constrained by the group-level estimate. The influence of the group level estimate on the participant-level estimate is inversely related to the uncertainty in the estimate: the smaller the uncertainty in the group estimate the larger its influence on the participant-level estimates. Importantly, the same holds for the other direction: the less data available for a certain participant in a certain condition the more that estimate is going to be constrained by the group parameter. Therefore, it is possible to estimate the model with only a single trial in a condition for a certain participant, although for this specific participant this specific estimate is going to be (almost) entirely determined by the group estimate. This can, for example, be mostly appreciated in Figure 4, where the participant-level estimates (i.e., the dots) for error trials (4B) are much more homogenous than those for correct trials (4A). This (likely) reflects the fact that much less error trials are available for each participant, and although it is still possible to estimate a group level estimate (because data are pooled across participants) the participant-level estimates are very much influenced by the group level estimate. We now added the following in the Materials and methods:

“The hierarchical Bayesian approach does not fit the model to individual subject’s data, but rather it jointly fits the data of the entire group. Therefore, data from participants with low trial counts in certain conditions does not contribute much to the posteriors for the respective condition. At the same time, participant-level estimates are estimated, but these are constrained by the group-level estimate. One obvious advantage of this approach is that participants with unequal trial numbers across conditions can contribute to the analysis, whereas in traditional approaches their data would be lost.”

5) The reviewers appreciate the authors' attempt to remove confounds caused by slow fluctuations of RT, accuracy and confidence. However, it is unclear that the correction procedure is adequate. Three related concerns are raised:

We thank the reviewers for raising this very important point. The comment suggests to us that the reviewers appreciate the necessity of factoring out slow drifts in overall performance. Note that in the previous version of our manuscript, we reported correlations between RT, confidence and accuracy on trial_n-1_ and trial_n_, which – we realized – can result from both slow drifts in performance and/or strategic trial-by-trial effects. Therefore, these have been removed and we now report the following on:

“Indeed, slow (‘scale-free’) fluctuations similar to those reported previously (Gilden, 2003; Palva et al., 2013) were present in the current RT and confidence rating time series as assessed by spectral analysis. Slopes of linear fits to log-log spectra were significant for both RTs, *b* = -.42, *t*(23) = -11.21, *p* <.001, and confidence, *b* = -.60, *t*(23) = -13.15, *p* <.001 (data not shown)”.

Upfront, we would like to solidify the intuition that some correction of slow performance drifts is, in fact, critical to assess the rapid, confidence-dependent bound updating effects we set out to test here. In a simulation shown in Author response image 5, we took the data from participant 1 in Experiment 1 and updated the bound based on the participant’s actual confidence ratings (high confidence: -.05, medium confidence: +.08, perceived error: +.2; note that these values were directly based on the fit of Experiment 1). Thus, all variations in decision bound are driven by confidence (left panel of Author response image 5).

**Author response image 5. respfig5:** 

When conditioning the decision bound from trial n+1 on confidence rating from trial n, this shows a dip in subsequent decision bound following low confidence (right upper panel: “uncorrected analysis”; high confidence *M* = 1.67, medium confidence *M* = 1.63, perceived errors *M* = 1.87), despite the fact that the bounds were forced to increase (by our simulation design) following medium confidence. In contrast, when using our approach of correcting for slow performance drifts (i.e. subtracting the bound conditioned on confidence on trial n+2), the resulting values closely match the ground truth (right lower panel: “correcting for n+2”; high confidence *M* = -.05, low confidence *M* =.08, perceived error *M* =.32). This indicates that the approach is appropriate for dealing with slow performance drifts.

However, we acknowledge that it comes at the cost of some ambiguity due to possible effects of decision bounds on subsequent confidence ratings (although there is no theoretical rationale for such an effect). We therefore now show that an alternative procedure for removing the slow performance drifts yields largely the same results as the approach used in the main paper and the above simulation (see our response to part b. of this comment).

We now add:

“If the decision bound is not fixed throughout the experiment (i.e., our data suggest that it is dynamically modulated by confidence), it is theoretically possible that specific levels of confidence judgments appear more frequently at specific levels of decision bound (e.g., everything else equal, high confidence trials are expected mostly in periods with a high decision bound).”

a) It is unclear that the underlying assumption of the analysis is correct. Of course confidence on trial n+2 cannot influence the decision bound on n+1, but could the decision bound on n+1 influence confidence on n+2? In both directions, these are just correlations, What support is there for the causal claim the authors are making of confidence on trial n changing bound on n+1?

We fully agree that claims about causality are not warranted given the correlative nature of our approach. We have now rephrased or toned down all statements that could be read as implying causality. This includes even a change in the title. That said, we would like to highlight that (i) our main conclusion does not depend on our specific approach for correcting for slow performance drifts (see below), and (ii) the temporal ordering of effects (confidence rating on trial n predicts bound on trial n+1) satisfies a commonly used (weak) criterion for causality.

b) The motivation of the analysis is to remove generic autocorrelations from the data (due to, e.g., long periods of high/low confidence correlated with low/high RT), under the assumption that trial n+2 confidence cannot be causally associated with trial n+1 decision bound. However, based on the unnormalized, "simple effects of confidence on decision bound," the effects seem to be mainly coming from trial n+2, rather than trial n (e.g., Figure 5—figure supplement 1, Figure 8—figure supplement 4, and Figure 5—figure supplement 3). This is fairly consistent in the behavioral data and is especially striking for the relation between Pe amplitude and decision bound. How does it affect whether we believe trial n+2 is an appropriate baseline measure? More important, these observations appear to challenge the main claim that confidence-dependent bound changes are shaped by trial n. Can the authors clarify why this challenge is not critical?

We have clarified in two independent ways that this challenge is not critical. First, for Experiments 1 and 3, the main results hold in the ‘raw measures’ without subtraction of trial n+2 confidence effects (Figure 3—figure supplement 2 and Figure 5—figure supplement 5). Only for Experiment 2 is the effect not quite significant (Figure 5—figure supplement 2; but note that such an absence of effect is hard to interpret given the simulation reported above). Second, we now show that our main conclusions remain unchanged when using a complementary approach to control for slow performance drifts: namely one that is analogous to the approach established in the post-error slowing literature. Here, the concern is that post-error and post-correct trials are not evenly distributed across the course of an experiment. Errors typically appear in the context of errors, a period also characterized by slow reaction times, hence creating an artificial link between errors on trial *n* and slow RTs on trial *n*+1. To eliminate this confound, Dutilh et al. (2012) proposed to compare post-error trials to post-correct trials that originate from the same locations in the time series. To achieve this, they examined performance (i.e., RT, accuracy) on trial *n+*1 when trial *n* was an error, compared to when trial *n* was correct *and* trial *n+*2 was an error (thus controlling for attentional state). We adapted this approach to our case (trial to trial variations in confidence ratings) as follows: similar as our previous approach we fitted a HDDM regression model to the data estimating subsequent decision bound as a function of decision confidence. However, we no longer added confidence on trial *n+*2 as a regressor, instead we only selected trials that originate from the same (“attentional state”) locations in the time series. Specifically, when comparing subsequent decision bound as a function of high confidence versus subjective errors, we compared the decision bound on trial *n+*1 when trial*n* was a perceived error, compared to when trial*n* was judged with high confidence *and* trial *n+*2 was a perceived error. In order to compare high confidence and low confidence, we compared the decision bound on trial *n+*1when trial *n* was judged with low confidence, compared to when trial*n* was judged with high confidence *and* trial *n+*2 was judged with low confidence. The results are shown in Figure 3—figure supplement 3, Figure 5—figure supplement 3 and 6, and Figure 8—figure supplement 4. In brief, using this approach we obtained largely the same findings as reported before, even when only selecting correct trials (there were not enough trials to only select errors).

The fact that we obtained largely the same findings using a different control further strengthens our confidence in our findings. After careful deliberation, we decided to keep the original analyses in the main paper and show the results from this complementary approach as controls in the supplementary materials. This is because our initial analysis allows to use all of the data, and fit these using a single model, making this a much more statistically powerful approach. By contrast, in the new approach a heavy sub-selection of trials is required, and a separate model needs to be fitted for each comparison. If the editors and reviewers prefer to see the results from the alternative approach in the main figures, we would be glad to move them there.

We now added the full explanation of this in the Materials and methods section:

“A possible concern is that the decision bound on trial_n+1_ affected confidence ratings on trial_n+2_, which would complicate the interpretation of the results of our approach. Thus, we also used a complementary approach controlling for slow drifts in performance, which is analogous to an approach established in the post-error slowing literature (Dutilh et al., 2012; Purcell and Kiani, 2016). In that approach, post-error trials are compared to post-correct trials that are also pre-error. As a consequence, both trial types appear adjacent to an error, and therefore likely stem from the same location in the Experiment. We adopted this approach to confidence ratings as follows: decision bound and drift rate on trial_n+1_ were fitted in separate models where i) we compared the effect of low confidence on trial_n_ to high confidence on trial_n_ for which trial_n+2_ was a low confidence trial, and ii) we compared the effect of perceived errors on trial_n_ to high confidence trials on trial_n_ for which trial_n+2_ was a perceived error. Thus, this ensured that the two trial types that were compared to each other stemmed from statistically similar environments. For the EEG data, we fitted a new model estimating decision bound and drift rate on trial_n+1_ when trial_n_ stemmed from the lowest Pe amplitude quantile, compared to when trial_n_ stemmed from the highest Pe amplitude quantile and trial_n+2_ stemmed from the lowest Pe amplitude quantile.”

And the results are referred to in the Results section:

“A possible concern is that the decision bound on trial_n+1_ affected confidence ratings on trial_n+2_, which would confound our measure of the effect of confidence on trial_n_ on decision bound on trial_n+1_. Two observations indicate that this does not explain our findings. First, the observed association between confidence ratings on trial_n_ and decision bound on trial_n+1_ was also evident in the “raw” parameter values for the bound modulation, that is, without removing the effects of slow performance drift (Figure 3—figure supplement 2). Second, when using a complementary approach adopted from the post-error slowing literature (Dutilh, Ravenzwaaij, et al., 2012; see Materials and methods), we observed largely similar results (see Figure 3—figure supplement 3).”

“Finally, we again observed a robust effect of confidence ratings on subsequent decision bound when using the above described alternative procedure to control for slow performance drift (Figure 5—figure supplement 3 and Figure 5—figure supplement 6). In Experiment 3 (Figure 5—figure supplement 5) but not in Experiment 2 (Figure 5—figure supplement 2), this effect was also present without controlling for slow performance drifts.”

“Similar findings were obtained using our alternative approach to control for slow performance drifts (Figure 8—figure supplement 4)”

c) The authors use the same model to quantify the effect of trial n or n+2 on trial n+1 and subtract the two effects to filter out slow fluctuations. This method would be successful only to the extent that the applied models are good at fitting the data. If a model fails to adequately capture the data, the residual error could contain slow fluctuations of behavior asymmetrically for the n/n+1 and n+2/n+1 analyses, which could contribute to results. There are reasons to be concerned because we do not know how well the DDM fits the behavior of individual participants, especially their RT distributions (not just the mean RT across the population; also consider large lapses and shallow RT functions). Showing residual errors of the model for RT and the autocorrelation of these residual errors would be useful to alleviate this concern.

As suggested, for each experiment we simulated data from the model and then subtracted the predicted RTs from the observed RTs in order to obtain residuals from the model. In Author response image 6 we show for each experiment that the residuals are not different between the different conditions (left) and that there is no autocorrelation in these residuals (right).

**Author response image 6. respfig6:** 

6) A point that deserves discussion and possibly looking into is the role of RT in determining the relationship between confidence and subsequent adjustment. From past papers, in varying-difficulty contexts, lower confidence can result from a decision reaching commitment at too low a level of cumulative evidence, OR alternatively from a crossing made later in the trial. The latter is more an indication that the current trial had weaker evidence than that the bound was not high enough. Setting the bound higher for the latter kind of trial would bring an increase in accuracy but at a vast cost in terms of time. Bound increases for low confidence do not clearly seem to be universally beneficial, and it would seem to depend on the cost of time, presence of deadlines etc. This kind of territory may be covered in papers like Meyniel et al., but either way, it seems a matter worth discussing in the current paper.

We agree this is an interesting point. We now touch upon this issue in the Discussion:

“Trial-to-trial variations in decision confidence likely result from several factors. For example, confidence might be low because of low internal evidence quality (i.e., low drift rate) or because insufficient evidence has been accumulated before committing to a choice (i.e., low decision bound). When the bound is low and results in low confidence, it is straightforward to increase the bound for the subsequent decision in order to improve performance. When drift rate is low, increasing the subsequent bound might increase accuracy only slightly, but at a vast cost in terms of response speed. Future work should aim to unravel to what extent strategic changes in decision bound differ between conditions in which variations in confidence are driven by a lack of accumulated evidence or by a lack of instantaneous evidence quality.”